# Cooperative chemoenzymatic and biocatalytic cascades to access chiral sulfur compounds bearing C(sp³)–S stereocentres

Fei Zhao [1], Ariane Mattana[1], Ruqaiya Alam [1], Sarah L. Montgomery [2], Akash Pandya[2], Fabrizio Manetti [3], Beatriz Dominguez[2] ✉ & Daniele Castagnolo [1] ✉

Biocatalysis has been widely employed for the generation of carbon-carbon/ heteroatom stereocentres, yet its application in chiral C(sp³)–S bond construction is rare and limited to enzymatic kinetic resolutions. Herein, we describe the enantioselective construction of chiral C(sp³)–S bonds through ene-reductase biocatalyzed conjugate reduction of prochiral vinyl sulfides. A series of cooperative sequential/concurrent chemoenzymatic and biocatalytic cascades have been developed to access a broad range of chiral sulfides, including valuable β-hydroxysulfides bearing two adjacent C(sp³)–S and C(sp³)–O stereocentres, in a stereoconvergent manner with good to excellent yields (up to 96%) and enantioselectivities (up to >99% ee). Notably, this biocatalytic strategy allows to overcome the long-standing shortcomings of catalyst poisoning and C(sp²)/C(sp³)–S bond cleavage faced in transition-metal-catalyzed hydrogenation of vinyl sulfides. Finally, the potential of this methodology is also exemplified by its broader application in the stereo-convergent assembly of chiral C(sp³)–N/O/Se bonds with good to excellent enantioselctivities.

The transition-metal (TM)-catalyzed asymmetric hydrogenation of multisubstituted alkenes with H₂ in the presence of various chiral ligands constitutes one of the most straightforward strategies for the generation of tertiary or quaternary C(sp³) stereocentres from prochiral sp² carbons[1–3]. However, while the efficient construction of chiral C(sp³)–C, C(sp³)–O, and C(sp³)–N bonds through asymmetric hydrogenation of carbon-substituted alkenes, enolates, and enamines has been well documented in literature[4–10], the asymmetric hydrogenation of vinyl sulfides is much less explored, even though this transformation offers a straightforward path for the conversion of prochiral C(sp²)–S into C(sp³)–S stereocentres, which are largely present in natural products[11] and pharmaceutical agents[12] (Fig. 1a). Only few methods converting vinyl sulfides into chiral sulfides bearing C(sp³)–S stereocentres have been reported to date, and they employ TM catalysts such as Cu, Ru, Rh, Ir complexes together with chiral phosphine

ligands[13–20]. These approaches show limitations since the strongly coordinating divalent sulfur atoms of substrates and products can poison the metal catalysts[21–26], thus hindering the reaction and leading in some cases to poor conversions (Fig. 1b). In addition, the competing side reaction of desulfurization via C(sp²)/C(sp³)–S bond hydrogenolysis is also problematic itself, and the desulfurized thiol side products formed can further poison the metal catalysts, limiting the efficiency of such transformations.

While the enzymatic synthesis of chiral sulfoxides has been well studied to date[27–34], by contrast, the biocatalytic synthesis of chiral sulfides bearing a stereocentre at C(sp³)–S bond has been only attempted by few research groups, through the enzymatic kinetic resolution (EKR) or dynamic kinetic resolution (DKR) of racemic sulfur substrates with lipases[35–37], 5-(hydroxymethyl)furfural oxidases (HMFO)[38], monoamine oxidases (MAO-N)[39], nitrilases[40], ketoreductases (KRED)[41] (Fig. 1c).

¹Department of Chemistry, University College London, London, UK. ²Johnson Matthey, Cambridge, UK. ³Department of Biotechnology, Chemistry and Pharmacy, University of Siena, Siena, Italy. ✉e-mail: beatriz.dominguez@matthey.com; d.castagnolo@ucl.ac.uk

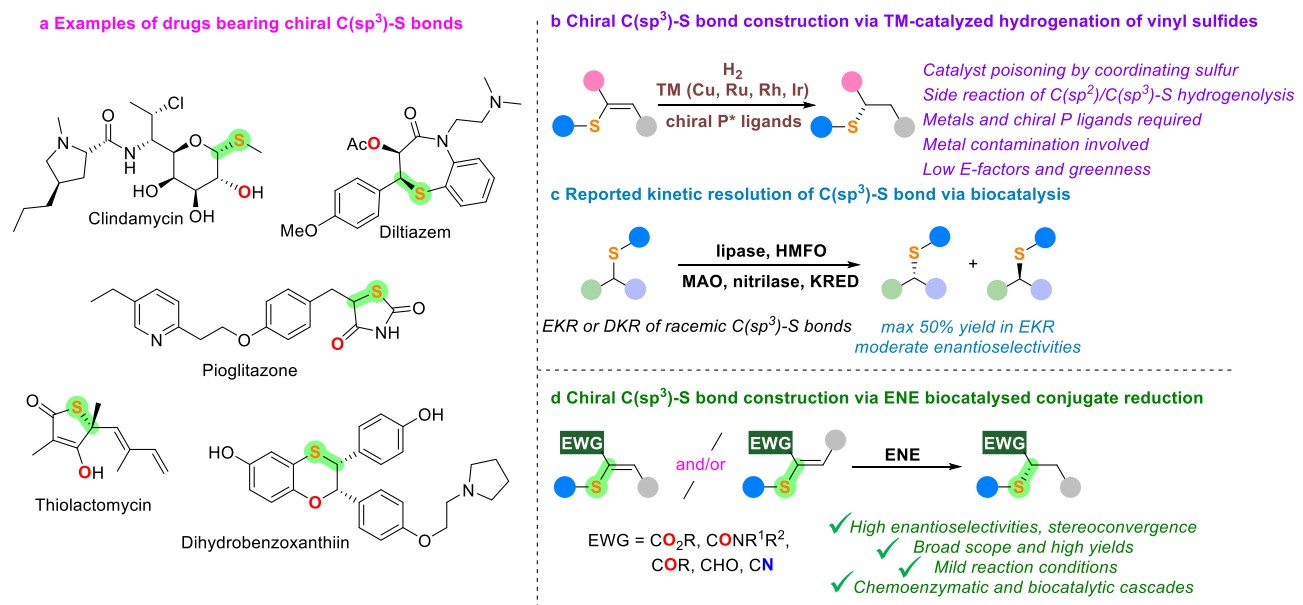

**Fig. 1 | Existing methods for chiral C(sp³)−S bond construction and our design.** **a** Representative natural and pharmaceutical products bearing chiral C(sp³)−S bonds. **b** Chiral C(sp³)−S bond construction via TM-catalyzed hydrogenation of vinyl sulfides. **c** Reported chiral C(sp³)−S bond construction via biocatalysis. **d** The design of this work: chiral C(sp³)−S bond construction via ENE biocatalyzed conjugate reduction.

However, these EKR and DKR methods do not generate C(sp³)−S stereocentres from prochiral C(sp²)−S bonds but rather they separate racemic mixtures of enantiomers, often resulting in poor yields (< 50% in EKR) and moderate enantioselectivities.

Following our interest in developing biocatalytic approaches towards enantiomerically pure chiral sulfur compounds[30,33,34,39–41], herein we report the biocatalytic synthesis of chiral sulfides through ene-reductase (ENE) catalyzed reduction of prochiral C(sp²)−S into stereodefined C(sp³)−S bonds (Fig. 1d). ENE biocatalysts have been successfully employed to enantioselectively synthesize a variety of sulfides bearing a stereocentre at C(sp³)−S bond in a stereoconvergent manner under mild conditions. Moreover, a series of one-pot cooperative sequential/concurrent chemoenzymatic and biocatalytic cascades have also been developed for the facile and more sustainable assembly of chiral sulfides bearing C(sp³)−S stereocentres. Finally, this methodology has also been extended to the construction of chiral C(sp³)−N/O/Se bonds as well as valuable β-hydroxysulfides bearing two adjacent C(sp³)−S and C(sp³)−O stereocentres through the combination of ENE and alcohol dehydrogenase (ADH) biocatalysts in cooperative biocatalytic cascades, highlighting the broad application and potential of this protocol.

## Results

### Reaction development and optimization
The identification of the best ENE biocatalyst was first conducted at an analytical scale with dimethyl 2-(phenylthio)fumarate (Z)-**1aa**, a vinyl sulfide bearing two electron-withdrawing groups (EWGs), as the model substrate (Table 1). Using D-glucose/glucose dehydrogenase GDH-101 as the cofactor recycling system, (Z)-**1aa** was treated with a panel of ENEs (cell-free extracts)[42–44] in the presence of NADP⁺ cofactor in a KPBS (pH 7.0)/DMSO (19/1) solution at 30 °C for 24 h (entries 1–7). With the exception of ENE-105, all other ENEs were able to catalyze the reduction of the carbon-carbon double bond efficiently. Remarkably, ENE-101 turned out to be the best biocatalyst, leading to (S)-**2aa** with full conversion (> 99%) and excellent enantioselectivity (> 99% ee). The absolute configuration of (S)-**2aa** was confirmed by comparing the optical rotation value of its LiAlH₄-reduced product with the one reported in literature[45]. With ENE-101 as the biocatalyst, GDH-5 and

GDH-8 were then tested as the cofactor regeneration enzymes. Both enzymes worked as efficiently as GDH-101, providing (S)-**2aa** in identical conversions and ee values (entries 8 and 9). ENE-101 also accepted NAD⁺ as the cofactor without affecting the conversion and ee value (entry 10). Increasing the reaction temperature to 37 °C was unbeneficial, resulting in a decrease in enantioselectivity (entry 11). Intriguingly, the reduction of the isomeric substrate (E)−**1aa** afforded the same enantiomer (S)-**2aa** in full conversion (> 99%) and good enantioselectivity (88% ee) (entry 12). Sulfide (S)-**2aa** was also obtained in a stereoconvergent manner from a Z/E (4.1/1) mixture of **1aa**, obtained from the reaction of thiophenol with dimethyl acetylenedicarboxylate (DMAD), with > 99% conversion and 97% ee (entry 13). Finally, (S)-**2aa** was synthesized at a preparative scale from (Z)−**1aa**, (E)−**1aa** or the Z/E (4.1/1) mixture in high isolated yields (91–95%) and good to excellent enantioselectivities (88- > 99% ee) (entries 1, 12 and 13). Blank experiments showed that ENE-101 was essential for the asymmetric reduction (entry 14). A time course experiment disclosed that (Z)-**1aa** is more reactive than (E)-**1aa** (Supplementary Table 3).

### Substrate scope of 2-thio fumarates and maleates
The substrate scope of the ENE-101 biocatalyzed conjugate reduction of a series of dimethyl 2-arylthio fumarates (Z isomers, (Z)−**1**) and maleates (E isomers, (E)−**1**), bearing two electron-withdrawing substituents on the double bond was first investigated (Fig. 2). The vinyl sulfides (Z)-**1ab-ah** and (E)-**1ab-ah** were both converted into the corresponding chiral sulfides (S)-**2ab-ah** in a stereoconvergent manner with good to excellent yields (50–96%) and enantioselectivities (72- > 99% ee). No obvious differences in reactivity and enantioselectivity were observed between the fumarate and maleate substrates. By contrast, the dimethyl 2-thio-fumarates (Z)-**1ai-al** bearing 4-MeO-Ph, 3,5-di-Me-Ph, naphthalen-1-yl or naphthalen-2-yl groups at the sulfur atom turned out to be more reactive than the corresponding maleates (E)-**1ai-al**. Although both the fumarate and maleate substrates afforded sulfides (S)-**2ai-al** in good to excellent enantioselectivities (79- > 99% ee), good to excellent conversions (83- > 99%) were only observed with the fumarate substrates. Remarkably, the stereoconvergent synthesis of (S)-**2ai-al** directly from the mixtures of the corresponding fumarates and maleates was achieved in moderate to good yields (36–86%) and

## Table 1 | Optimization of reaction conditions[a]

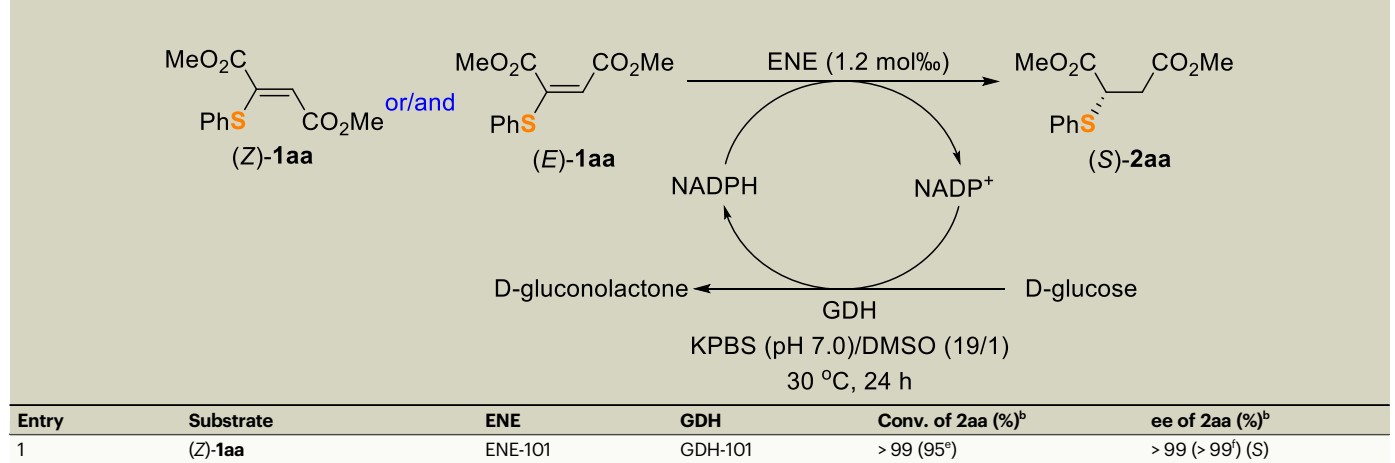

| Entry | Substrate | ENE | GDH | Conv. of 2aa (%)[b] | ee of 2aa (%)[b] |
|---|---|---|---|---|---|
| 1 | (Z)-1aa | ENE-101 | GDH-101 | > 99 (95[e]) | > 99 (> 99[f]) (S) |
| 2 | (Z)-1aa | ENE-102 | GDH-101 | > 99 | 93 (S) |
| 3 | (Z)-1aa | ENE-103 | GDH-101 | > 99 | 93 (S) |
| 4 | (Z)-1aa | ENE-105 | GDH-101 | 82 | 47 (S) |
| 5 | (Z)-1aa | ENE-107 | GDH-101 | > 99 | 93 (S) |
| 6 | (Z)-1aa | ENE-108 | GDH-101 | > 99 | 93 (S) |
| 7 | (Z)-1aa | ENE-109 | GDH-101 | > 99 | 92 (S) |
| 8 | (Z)-1aa | ENE-101 | GDH-5 | > 99 | > 99 (S) |
| 9 | (Z)-1aa | ENE-101 | GDH-8 | > 99 | 99 (S) |
| 10[c] | (Z)-1aa | ENE-101 | GDH-101 | > 99 | > 99 (S) |
| 11[d] | (Z)-1aa | ENE-101 | GDH-101 | > 99 | 91 (S) |
| 12 | (E)-1aa | ENE-101 | GDH-101 | > 99 (93[e]) | 88 (88[f]) (S) |
| 13 | (Z)-1aa/(E)-1aa (4.1/1) | ENE-101 | GDH-101 | > 99 (91[e]) | 97 (96[f]) (S) |
| 14 | (Z)-1aa | / | GDH-101 | 0 | / |

[a]Reaction conditions: (Z)-1aa and/or (E)-1aa (0.02 mmol), ENE (5.0 mg, 1.2 mol‰), GDH (2.5 mg), D-glucose (0.1 mmol), NADP⁺ (1 μmol), 950 μL KPBS (250 mM, pH 7.0), DMSO (50 μL), 30 °C, 24 h.
[b]Determined by chiral HPLC analysis. [c]NAD⁺ was used instead of NADP⁺ as the cofactor. [d]The reaction was performed at 37 °C. [e]Isolated yield of **2aa** when 0.1 mmol of **1aa** was used. [f]The ee value of **2aa** when 0.1 mmol of **1aa** was used. ENE-101 (65.8 U/mg), ENE-102 (5.4 U/mg), ENE-103 (3.0 U/mg), ENE-105 (9.0 U/mg), ENE-107 (101.0 U/mg), ENE-108 (6.0 U/mg), ENE-109 (6.1 U/mg), GDH-101 (32.0 U/mg), GDH-5 (51.2 U/mg), GDH-8 (1.6 U/mg).

excellent enantioselectivities (96- > 99% ee). On the contrary, both dimethyl 2-thio-fumarate (Z)-**1am** and maleate (E)-**1am** bearing a furan-2-yl group at the sulfur atom were converted into (S)-**2am** in good to high yields (80–91%), but an excellent enantioselectivity (99% ee) was only observed with the former. Interestingly, the reactions of dimethyl 2-thio-fumarates (Z)-**1an-ar** bearing an alkyl group at the sulfur atom afforded (S)-**2an-ar** in excellent conversions (> 99%), yields (90–95%) and enantioselectivities (94- > 99% ee), while the reactions of the corresponding maleates (E)-**1an-ar** produced the opposite enantiomers (R)-**2an-ar** in poor conversions (20–45%) and moderate enantioselectivities (50–66% ee). Regardless of the difference in conversions, the reactions of both dimethyl 2-thio-fumarate and maleate bearing a cyclohexyl group on the sulfur atom provided (S)-**2as** in high enantioselectivities (90- > 99% ee). In particular, dimethyl 2-thio-maleate (E)-**1at** was well tolerated, affording (S)-**2at** in good yield (51%) and enantioselectivity (62% ee). Then, a series of fumarates (Z)-**1au-aw** and maleates (E)-**1au-aw** bearing diethyl, diisopropyl, and di-*tert*-butyl substituents on the ester moiety were evaluated. Except for diethyl 2-phenylthio-fumarate, which reacted smoothly to give (S)-**2au** in 90% yield and 99% ee, the reactions of other substrates furnished products (S)-**2au-aw** in poor conversions (≤ 27%). This may be attributed to the steric hindrance caused by the bulky isopropyl and *tert*-butyl ester groups. Pleasingly, the 3-thiomaleimide reacted well to provide (S)-**2ax** in good yield (89%) and enantioselectivity (69% ee). Remarkably, the dimethyl 2-sulfonyl fumarates (Z)-**1ay-az** were also suitable substrates, providing chiral sulfones (S)-**2ay-az** in good yields (80–82%) and ee

values (89–90%). Moreover, the ENE biocatalyzed conjugate reduction was also compatible with dimethyl 2-(phenylamino)maleate (E)-**1ba**, which was converted into chiral amine (S)-**2ba** in > 99% ee, albeit with low conversion (24%). Similarly, chiral ethers (S)-**2bb-bc** were synthesized stereoconvergently from inseparable mixtures of dimethyl 2-oxy-fumarates and maleates with excellent enantioselectivities (> 99% ee). Lastly, the reactions of dimethyl 2-phenylselanyl fumarate, maleate, and their mixtures were also investigated, providing chiral selenide (S)-**2bd** in 70–78% yields and 60-99% ee values.

### Chemoenzymatic cascades to access chiral sulfides

With the aim to make the synthetic protocol more convenient and greener, the possibility of developing a cooperative two-step one-pot chemoenzymatic cascade was then investigated, potentially allowing the direct synthesis of enantiopure sulfides **2** from commercially available thiophenols/thiols and DMAD (Fig. 3a). Delightfully, thiophenol reacted smoothly with DMAD in a KPBS (pH 7.0)/DMSO (18/1) solution at 30 °C for 24 h, leading to its full conversion into the corresponding isomeric mixture of vinyl sulfides (Z)-**1aa** and (E)-**1aa**, which were then reduced by ENE-101 added in one-pot to give (S)-**2aa** in stereoconvergent manner, good overall yield (65%) and good enantioselectivity (92% ee) (Supplementary Table 5). The general scope of this two-step one-pot chemoenzymatic cascade was then explored. Various thiophenols/thiols, as well as phenylselenol, underwent this cascade successfully to give (S)-**2ab-an** and (S)-**2bd** in 35–88% yields and 82–99% ee values. Noteworthily, no obvious loss in

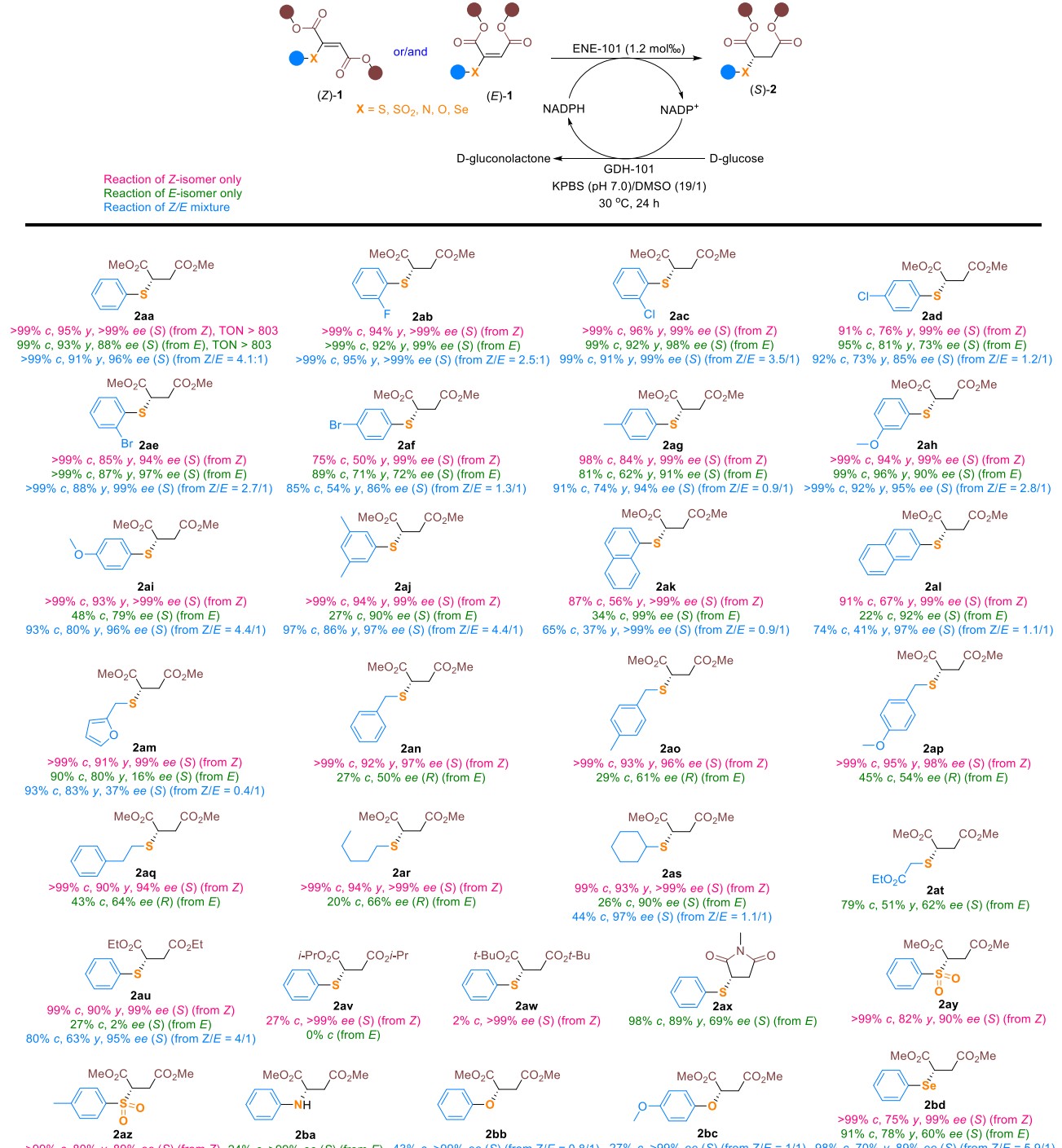

**Fig. 2 | Scope of fumarate and maleate substrates.** Reaction conditions: (*Z*)-**1** and/or (*E*)-**1** (0.1 mmol), ENE-101 (25 mg, 65.8 U/mg, 1.2 mol‰), GDH-101 (12.5 mg, 32.0 U/mg), D-glucose (0.5 mmol), NADP⁺ (5 μmol), 1.9 mL KPBS (250 mM, pH 7.0), DMSO (100 μL), 30 °C, 24 h. *c* = conversion; *y* = isolated yield; *ee* = enantiomeric excess. Isolated yields were not determined when the conversions were less than 50%. Turnover number (TON) was calculated as the moles of substrate that were converted by per mole of ENE-101 biocatalyst.

enantioselectivity was observed as compared with the direct reduction of the corresponding vinyl sulfides/selenide, suggesting the efficiency of the two-step one-pot cascade.

Furthermore, we developed a more convenient cooperative one-step one-pot chemoenzymatic cascade, in which thiophenols/thiols, DMAD, ENE-101, NADP⁺, and the cofactor recycling system (D-glucose/GDH-101) were all added together, allowing a more straightforward synthesis of chiral sulfides in a single operation (Fig. 3b). The main challenges of such cascade were represented by the possibility that the nucleophilic thiophenols/thiols and electrophilic DMAD may interact and inactivate the ENE-101 and GDH-101 enzymes by covalent bonding as well as by the mutual compatibility of all the components in this reaction system. Remarkably, the conceived cascade worked well on the model reaction, affording **2aa** with 56% yield and 92% ee in a KPBS (pH 7.0)/DMSO (19/1) solution at 30 °C for 24 h (Supplementary Table 7). As a result, a series of isomeric mixtures of vinyl sulfide

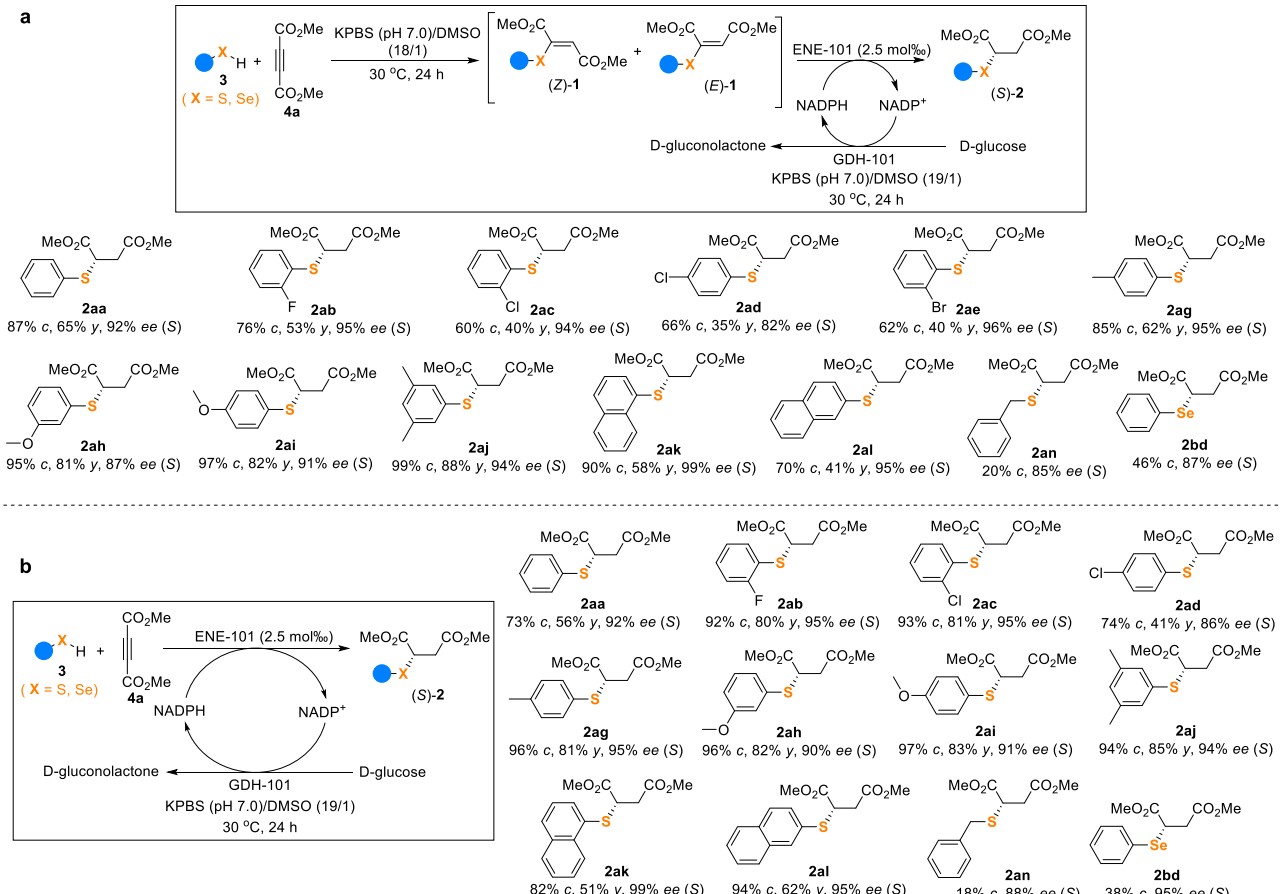

**Fig. 3 | Scope of the chemoenzymatic cascades for chiral sulfide synthesis.**
**a** Scope of the two-step one-pot chemoenzymatic cascade. Reaction conditions: **3** (0.1 mmol), **4a** (0.1 mmol), 3.6 mL KPBS (250 mM, pH 7.0), DMSO (200 µL), 30 °C, 24 h; then ENE-101 (50 mg, 65.8 U/mg, 2.5 mol‰), GDH-101 (25 mg, 32.0 U/mg), D-glucose (1 mmol), NADP⁺ (10 µmol) in 200 µL KPBS (250 mM, pH 7.0), 30 °C, 24 h.
**b** Scope of the one-step one-pot chemoenzymatic cascade. Reaction conditions: **3** (0.1 mmol), **4a** (0.1 mmol), ENE-101 (50 mg, 65.8 U/mg, 2.5 mol‰), GDH-101 (25 mg, 32.0 U/mg), D-glucose (1 mmol), NADP⁺ (10 µmol), 3.8 mL KPBS (250 mM, pH 7.0), DMSO (200 µL), 30 °C, 24 h. c = conversion; y = isolated yield; ee = enantiomeric excess. Isolated yields were not determined when the conversions were less than 50%.

intermediates, formed via the addition of thiophenols/thiols to DMAD, were reduced in situ by ENE-101 in a stereoconvergent way to afford sulfides (S)-**2ab-an** in 41–85% yields and 86-99% ee values. Interestingly, selenide (S)-**2bd** was also obtained with a high ee value (95%) directly from phenylselenol. Again, no significant loss in enantioselectivity was observed.

## Mechanistic investigations

To gain some insights into the ENE-101 biocatalyzed reaction mechanism and stereochemistry, a series of deuterium-labeling experiments were performed (Fig. 4a). At first, treatment of (Z)-**1aa** under standard conditions with deuterated NADPH (NADPD), which was regenerated in situ by GDH-101 and D-glucose-$d_{12}$, resulted in 74% deuteration at the β position of the sulfur atom of product (S)-**2aa**-$d_1$. On the opposite, 96% deuterium incorporation at the α position of the sulfur atom of the product (S)-**2aa**-$d_{1'}$ was observed when (Z)-**1aa** was reacted in deuterated KPBS under optimal conditions. As expected, high deuterations at both α and β positions of the sulfur atom of the product (S)-**2aa**-$d_2$ were observed when (Z)-**1aa** was treated with NADPD in deuterated KPBS. These results suggest that the ester group linked to the prochiral carbon behaves as the activating EWG, and the first hydride from the ENE FMNH₂ cofactor attacks the less hindered carbon. According to the accepted mechanism for ENE conjugate reduction[46], an addition of the FMNH₂ cofactor hydride and a proton from a tyrosine residue generally occurs. Based on this, molecular docking calculations were performed to find the possible

binding pose of both (Z)-**1aa** and (E)-**1aa** within the catalytic pocket of the enzyme. For this purpose, an ENE-101 homology model was built from the NerA reductase from *Agrobacterium radiobacter* (PDB ID: 4JIC). (Z)-**1aa** was found in a binding mode with the ester moiety at the prochiral carbon forming a complex network of hydrogen bond interactions with the conserved donor residues His181 and Asn184, as well as Tyr186, which is usually responsible for the proton transfer (Fig. 4b, left). An additional anchor point was represented by a hydrogen bond between Tyr357 and the other ester group at the non-prochiral carbon. Interestingly, a similar binding mode was found for (E)-**1aa** (Fig. 4b, right), whose pose was rotated by about 90 degrees anti-clockwise in comparison with the binding pose of (Z)-**1aa**. The hydrogen bond network with both Asn184 and Tyr186 was maintained, while the interaction with His181 was missing. The similar binding mode of (Z)-**1aa** and (E)-**1aa** within the catalytic pocket accounts for the stereoconvergent generation of (S)-**2aa**.

## Substrate scope of α/β-arylthio-electron-deficient alkenes

A variety of electron-deficient alkenes **5** and **6** bearing a single EWG and sulfur substituent at the α or β position of the EWG were then tested as substrates for ENE-101 biocatalyzed conjugate reduction (Fig. 5a). At first, α-phenylthio vinyl methyl ketone (Z)-**5aa** was fully converted into the desired product (S)-**7aa** with good yield (90%) and enantioselectivity (72% ee) by ENE-101 in 6 h. A screening of the panel of ENEs on (Z)-**5aa** revealed that ENE-101 was still the best enzyme for this reduction (Supplementary Table 9). Shortening the reaction time to 1.5 h proved

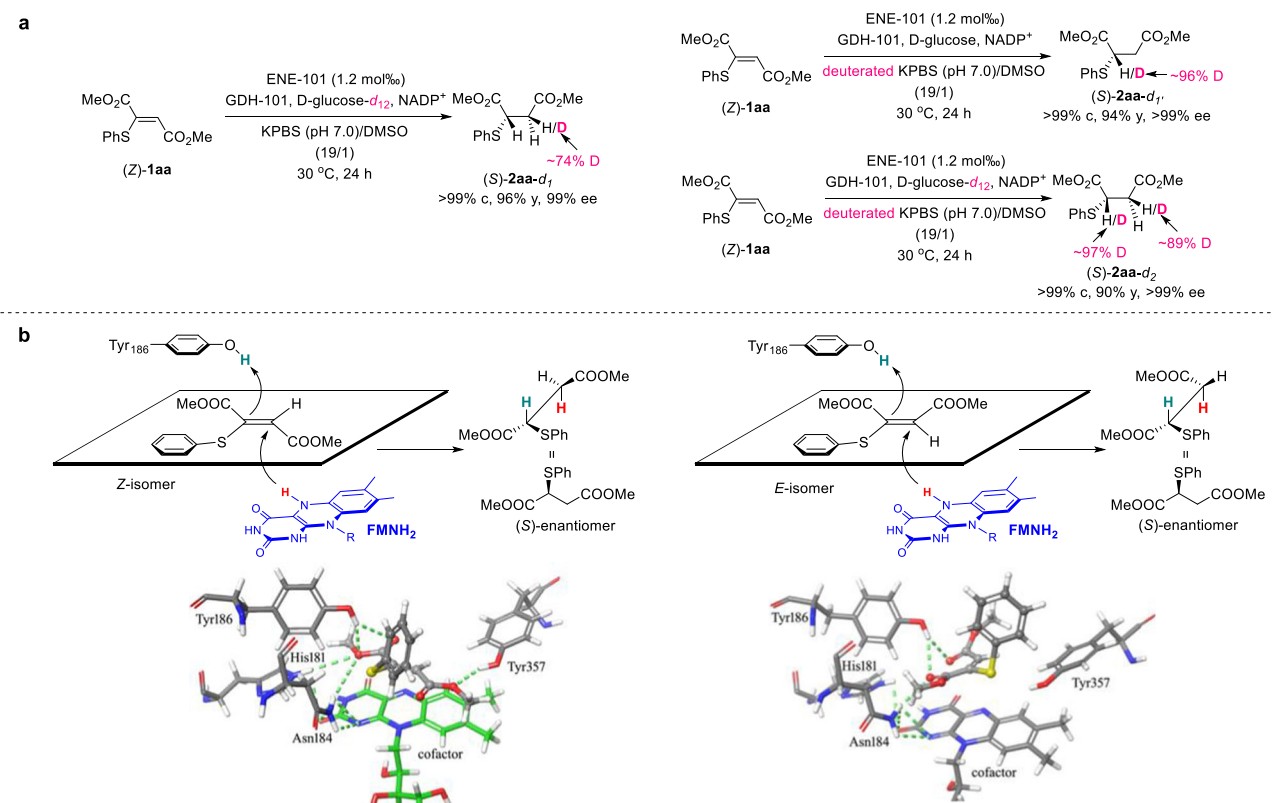

**Fig. 4 | Mechanistic Studies. a** Deuterium-labeling experiments. **b** Graphical representation of the best docked binding pose of (*Z*)-**1aa** and (*E*)-**1aa**. For clarity, only a few amino acids involved in the interactions with the substrates were represented, together with the cofactor.

to be beneficial for the biotransformation, leading to a remarkable improvement in the enantioselectivity (90% ee) without the loss in yield (91%). Such improvement is due to the configurational instability and fast racemization of (*S*)-**7aa** by keto-enol tautomerization in KPBS (Supplementary Table 10). However, it is important to note that (*S*)-**7aa** is configurationally stable when isolated from the reaction system. A set of vinyl methyl ketones (*Z*)-**5ab-af** bearing diverse sulfur substituents at the α position were then converted into (*S*)-**7ab-af** in good to high yields (72−92%) and enantioselectivities (90- > 99% ee) within 3 h. The reactions of α-thio vinyl methyl ketones (*Z*)/(*E*)-**5ag** and (*Z*)/(*E*)-**5ah** possessing a phenyl group on the double bond gave respectively (*S*)-**7ag** and (*S*)-**7ah** in 30−89% yields and 70-88% ee values. By contrast, α-phenylthio vinyl phenyl ketone (*Z*)-**5ai** cannot be reduced under standard conditions even after a prolonged time. Besides, α-phenylthio vinyl aldehyde (*Z*)-**5aj** could also be transformed into (*S*)-**7aj** but with poor ee values, likely due to the low enantioselectivity of the reaction itself and the in situ racemization of the product (Supplementary Table 12)[41,47,48]. While the reduction of α-phenylthio vinyl oxime **5ak** into **7ak** failed, the reaction of a *Z/E* (1/1) mixture of α-phenylthio vinyl nitrile **5al** proceeded smoothly to deliver (*S*)-**7al** in high yield (92%) and enantioselectivity (98% ee). Likewise, α-phenylthio vinyl ester (*Z*)-**5am** was well tolerated, affording (*S*)-**7am** in good yield (88%) and excellent enantioselectivity (> 99% ee). Surprisingly, this reaction was incompatible with β-thio vinyl ketone **6aa** or ester **6ab** as no or negligible formation of the desired products **8aa** or **8ab** was observed. These findings suggest that the sulfur substituents located at the α rather than β position of the EWG of electron-deficient alkenes were essential for the transformation. Fukui functions and atomic Fukui indices were used to describe the local chemical reactivity of α- and β-thio vinyl ketones/esters. Among the olefinic carbons, the highest positive values of the f_NN Fukui index for the lowest unoccupied molecular orbital (LUMO) are associated with

atoms most prone to a nucleophilic attack. Therefore, a comparison of the Fukui indexes calculated for the alkene substrates can explain the different reactivity. The α-thio vinyl ketone/ester (*Z*)-**5aa** and (*Z*)-**5am**, which were fully reduced, showed f_NN Fukui indexes for the LUMO of the β olefinic carbons of 0.425 and 0.432, respectively (Fig. 5b). On the contrary, the indexes calculated for the β olefinic carbons of (*Z*)-**6aa** and (*Z*)-**6ab**, which were barely or not reduced, were only 0.292 and 0.290, respectively, suggesting an electronic impact of the sulfur atom on the electrophilicity of β olefinic carbon. In addition, we cannot exclude that steric factors may also contribute to the different reactivity of α- and β-thio vinyl ketones/esters.

## Hydrogen-borrowing cascade to access chiral sulfides

A convenient and simple way for the synthesis of chiral sulfides **7** from **9** was then investigated through a one-step one-pot hydrogen-borrowing cascade (HBC)[49–52], in which only a catalytic amount of NADP[+] cofactor was used, and no cofactor recycling system (D-glucose/GDH-101) was employed (Fig. 5c). We assumed alcohols **9** could be oxidized by alcohol dehydrogenase (ADH) in the presence of NADP[+] to form vinyl ketone intermediates **5**, which could be reduced by ENE biocatalyst using the NADPH cofactor formed through the transfer of a hydride from **9** to NADP[+]. As a result, the combination of two enzymes, namely ADH-159, selected through the screening of a series of ADH enzymes on **9aa** (Supplementary Table 13), and ENE-101, achieved this HBC through relay catalysis, yielding **7aa** and **7ae** as examples in good enantioselectivities (74−88% ee), although the conversions were not satisfactory. The difference between the free energies of reactant **9aa** and product **7aa** was calculated according to the group-contribution method previously described in literature[53,54], and it turned to be about 14 kcal/mol, which is large enough to drive the reaction towards the formation of **7aa** before the system can reach equilibrium.

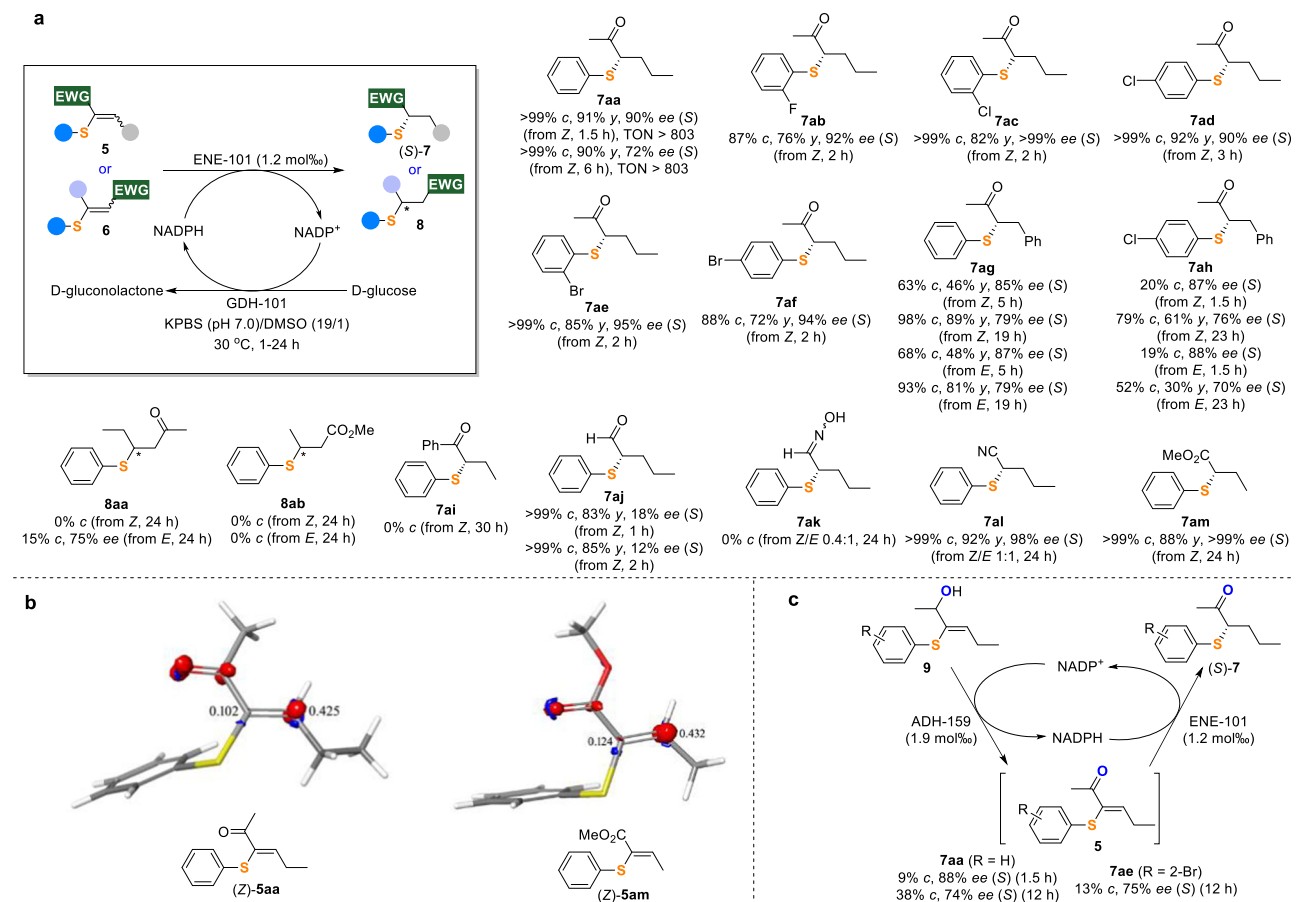

**Fig. 5 | Scope of α/β-thio electron-deficient alkenes and Fukui function analysis.**
**a** Scope of α/β-thio electron-deficient alkenes. Reaction conditions: **5** or **6**
(0.1 mmol), ENE-101 (25 mg, 65.8 U/mg, 1.2 mol‰), GDH-101 (12.5 mg, 32.0 U/mg),
D-glucose (0.5 mmol), NADP⁺ (5 μmol), 1.9 mL KPBS (250 mM, pH 7.0), DMSO
(100 μL), 30 °C, 1–24 h. TON was calculated as the moles of substrate that were
converted by per mole of ENE-101 biocatalyst. **b** Graphical representation of Fukui

functions and indexes for (Z)-**5aa** and (Z)-**5am**. **c** Scope of the hydrogen-borrowing
cascade. Reaction conditions: **9** (0.02 mmol), ENE-101 (5 mg, 65.8 U/mg, 1.2 mol‰),
ADH-159 (5 mg, 0.2 U/mg, 1.9 mol‰), NADP⁺ (1 μmol), 0.95 mL KPBS (250 mM, pH
7.0), DMSO (50 μL), 30 °C, 1.5 or 12 h. c = conversion; y = isolated yield; ee =
enantiomeric excess. Isolated yields were not determined when the conversions
were less than 50%. *Absolute configuration not assigned.

## Chemoenzymatic/biocatalytic cascades to access β-hydroxysulfides

The enantioselective synthesis of valuable β-hydroxysulfides **10** bearing
two adjacent stereocentres at the C(sp³)−S and C(sp³)−O bonds from
vinyl ketones **5** was then investigated through a two-step one-pot cas-
cade employing ENE-101 in tandem with a second reducing agent such
as NaBH₄ or ADH (Fig. 6a). At first, the ENE-101/NaBH₄ biocatalytic-
chemical (bio-chemo) cascade led to the synthesis of hydroxysulfides
**10aa**, **10ad** and **10ae** in good yields (74–86%) with good to excellent
enantioselectivities (84->99% ee) for both the *syn*- and *anti*-diaster-
eoisomers (around 4:1 *syn/anti*). The *syn* and *anti* diastereoisomers of
compounds **10** were assigned according to the literature[55]. Later, ADH-
153 and ADH-19, which were selected through the screening of a panel
of ADH enzymes on **7ad** (Supplementary Table 15), were used to replace
NaBH₄. The ENE-101/ADH-153 biocatalytic-biocatalytic (bio-bio) cas-
cade, in which the two enzymes shared the same cofactor and cofactor
recycling system, provided *syn*-(2S,3S)-**10aa-ae** exclusively with excel-
lent diastereoselectivities (100:0 *syn/anti*) and enantioselectivities
(> 99% ee) in all cases, thereby reflecting the higher efficiency of the bio-
bio over the bio-chemo cascade. Remarkably, when ADH-153 was
replaced by ADH-19 in the bio-bio cascade, the tandem reactions
overall gave the opposite *anti*-diastereoisomers (2R,3S)-**10aa-ae** as the
major products with 95->99% ee values.

Furthermore, a more convenient one-step one-pot ENE-101/ADH-
153 or ENE-101/ADH-19 bio-bio cascade for the synthesis of β-

hydroxysulfides **10** from **5** was attempted (Fig. 6b). The main chal-
lenge of such cascade is to certify ADH-153 or ADH-19 preferentially
reduces the ketone group of the α-thio ketone intermediates gener-
ated by ENE-101 catalyzed conjugate reduction of **5** rather than that of
the more abundant materials **5**. In fact, the reduction of the ketone
group of **5** by ADH-153 or ADH-19 would result in a non-conjugated
C = C bond which could not be reduced by ENE-101, thus terminating
the cascade. Notably, the designed one-step one-pot ENE-101/ADH-153
bio-bio cascade produced *syn*-(2S,3S)-**10aa-ae** in moderate to good
yields (38-83%) with excellent diastereoselectivities (98/2-100/0
*syn/anti*) and enantioselectivities (93->99% ee). By contrast, the one-
step one-pot ENE-101/ADH-19 cascade tended to give the opposite
*anti*-diastereoisomers as exemplified by (2R,3S)-**10ab-ad** as the major
products with >99% ee values. Noteworthily, as demonstrated by HPLC
analysis, the side reaction of carbonyl reduction of **5** by ADH-153 or
ADH-19 was almost negligible in this cascade, making this one-step
one-pot cascade equally efficient but simpler as compared with the
corresponding two-step one-pot cascade on the whole.

Finally, the synthesis of β-hydroxysulfides **10** from **9** was also
accomplished via a more challenging three-step one-pot bio-bio-bio
cascade (Fig. 7). Alcohols **9** were oxidized by ADH in the presence of
NADPH oxidase (NOX) and air into the corresponding α-thio vinyl
ketone intermediates **5**, which were subsequently reduced by the
addition of ENE-101 and D-glucose/GDH-101 to give α-thio ketone
intermediates **7**. The thio-ketones **7** were finally reduced by the same

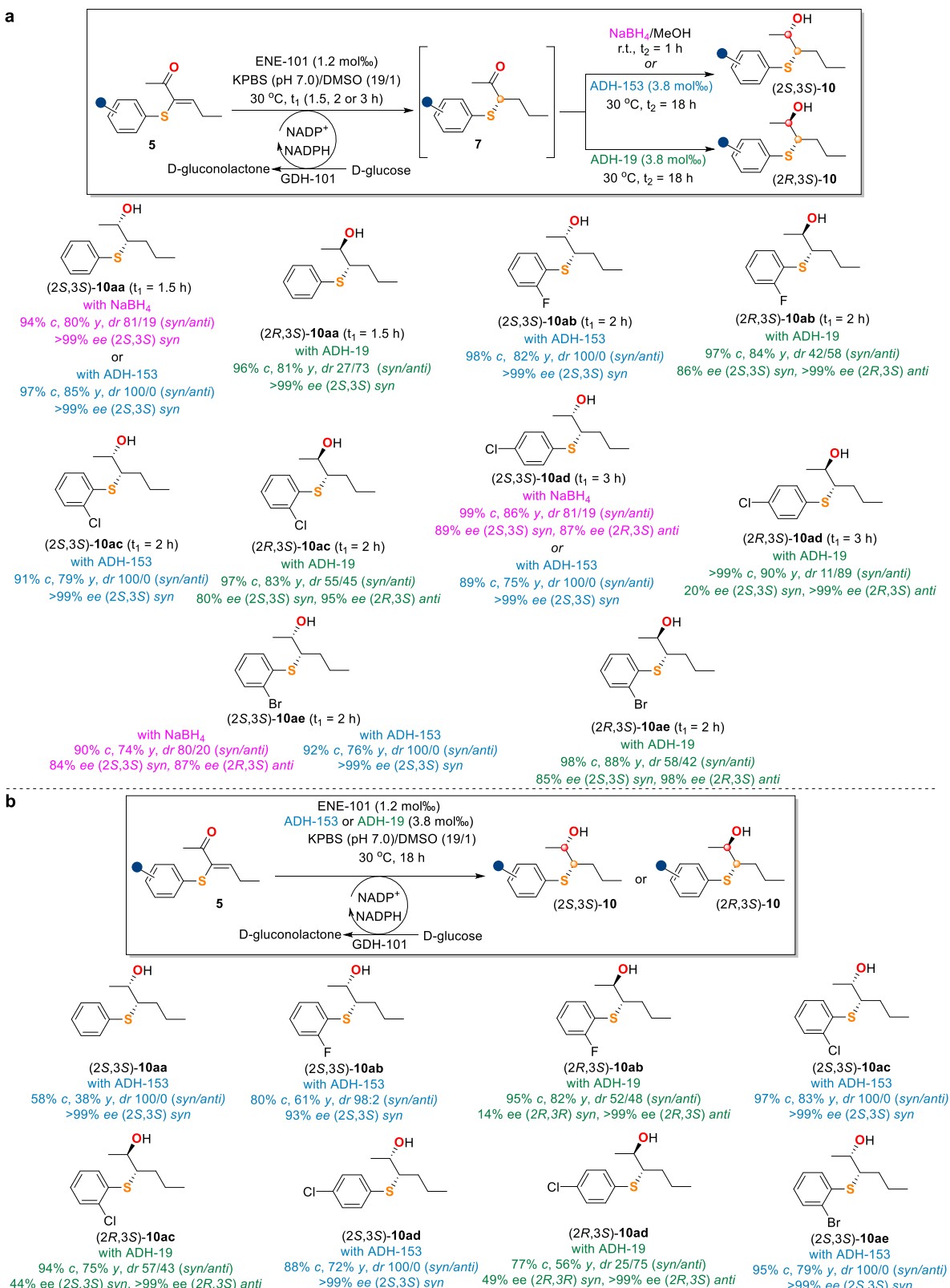

**Fig. 6 | Scope of bio-chemo/bio-bio cascades for β-hydroxysulfide synthesis.**
**a** Scope of the two-step one-pot bio-chemo/bio-bio cascade. Reaction conditions: **5** (0.1 mmol), ENE-101 (25 mg, 65.8 U/mg, 1.2 mol‰), GDH-101 (12.5 mg, 32.0 U/mg), D-glucose (0.5 mmol), NADP⁺ (5 μmol), 1.9 mL KPBS (250 mM, pH 7.0), DMSO (100 μL), 30 °C, 1.5, 2 or 3 h; then MeOH (1 mL), NaBH₄ (0.5 mmol), room temperature, 1 h; or ADH-153 (50 mg, 3.3 U/mg, 23.8 mol‰), 30 °C, 18 h; or ADH-19 (50 mg, 2.8 U/mg, 3.8 mol‰), 30 °C, 18 h. **b** Scope of the one-step one-pot bio-bio

cascade. Reaction conditions: **5** (0.1 mmol), ENE-101 (25 mg, 65.8 U/mg, 1.2 mol‰), ADH-153 (50 mg, 3.3 U/mg, 3.8 mol‰) or ADH-19 (50 mg, 2.8 U/mg, 3.8 mol‰), GDH-101 (12.5 mg, 32.0 U/mg), D-glucose (0.5 mmol), NADP⁺ (5 μmol), 1.9 mL KPBS (250 mM, pH 7.0), DMSO (100 μL), 30 °C, 18 h. *c* = conversion; *y* = overall isolated yield; *dr* = diastereomeric ratio; *ee* = enantiomeric excess. Isolated yields were not determined when the conversions were less than 50%.

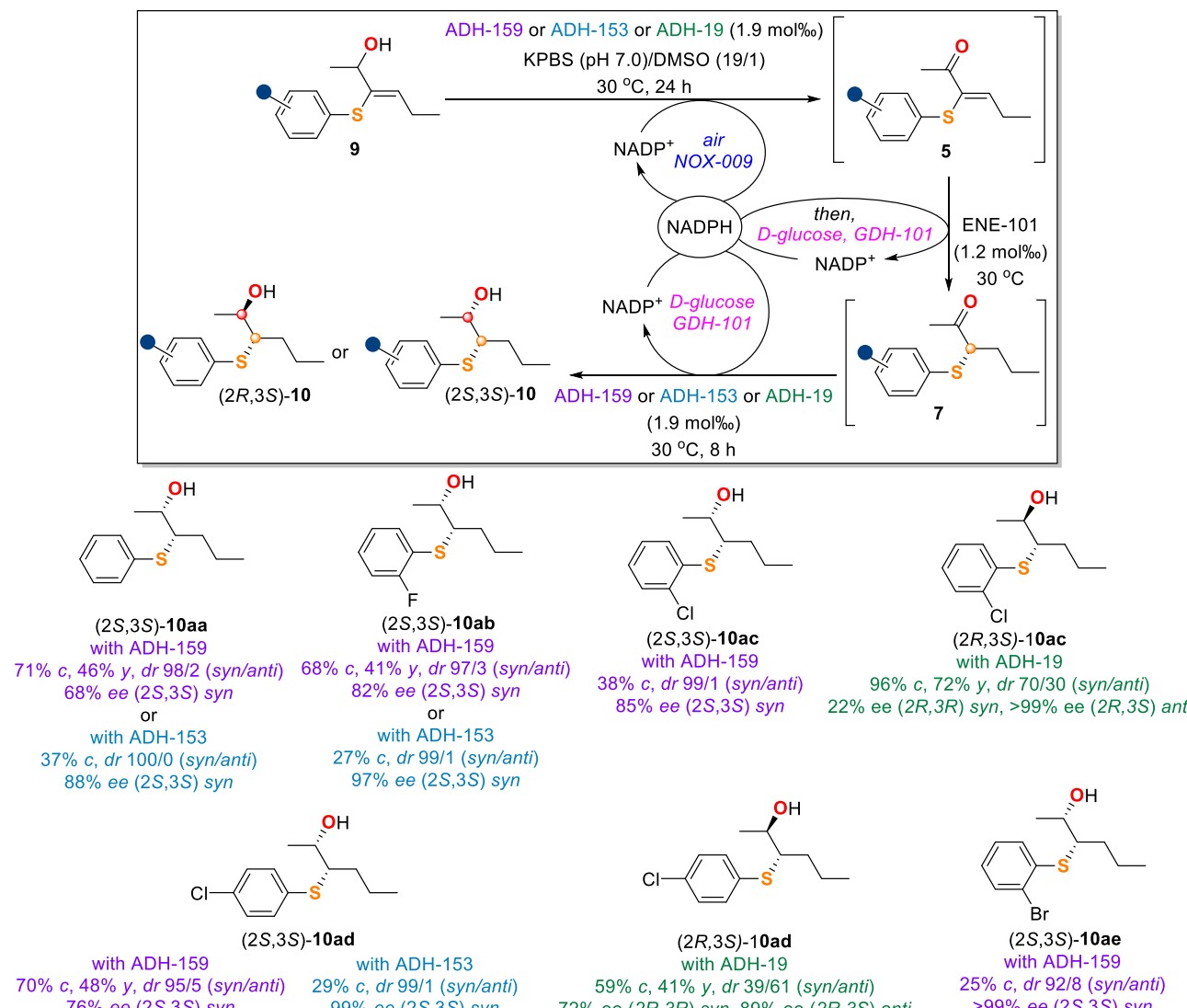

**Fig. 7 | Scope of three-step one-pot bio-bio-bio cascade for β-hydroxysulfide synthesis.** Reaction conditions: **9** (0.1 mmol), ADH-159 (25 mg, 0.2 U/mg, 1.9 mol‰) or ADH-153 (25 mg, 3.3 U/mg, 1.9 mol‰) or ADH-19 (25 mg, 2.8 U/mg, 1.9 mol‰), NOX-009 (25 mg), NADP⁺ (5 μmol), 3.8 mL KPBS (250 mM, pH 7.0), DMSO (200 μL), air, 30 °C, 24 h; then ENE-101 (25 mg, 65.8 U/mg, 1.2 mol‰), GDH-101 (12.5 mg, 32.0 U/mg), D-glucose (0.5 mmol), 30 °C, 8 h. *c* = conversion; *y* = overall isolated yield; *dr* = diastereomeric ratio; *ee* = enantiomeric excess. Isolated yields were not determined when the conversions were less than 50%.

ADH enzyme to provide the desired β-hydroxysulfides **10**. As a result, the cascade with ADH-159 or ADH-153 provided *syn*-(2*S*,3*S*)-**10aa-ae** with excellent diastereoselectivities (92/8-100/0 *syn/anti*) and good to excellent enantioselectivities (68- > 99% ee). The cascade with ADH-19 delivered the opposite *anti*-diastereoisomers as exemplified by (2*R*,3*S*)-**10ac** and (2*R*,3*S*)-**10ad** as the major products with 89- > 99% ee values. The unsatisfactory conversions observed in some cases were mainly due to the incomplete consumption of **9**. Of note, the ADH enzyme used in this cascade acts as a bifunctional biocatalyst, firstly oxidizing alcohols **9** and then reducing intermediates **7**.

## Discussion

In summary, we report the enantioselective synthesis of chiral sulfur compounds through ENE biocatalyzed construction of chiral C(sp³)−S bonds from prochiral vinyl sulfides. ENE-101 biocatalyst can reduce stereoconvergently *Z/E* mixtures of vinyl sulfides into enantiopure sulfides with good to excellent enantioselectivities and yields. We found that a sulfur substituent located at the α- rather than β-position of the EWG of vinyl sulfides is essential for this biotransformation. A

series of cooperative sequential/concurrent chemoenzymatic and biocatalytic cascades, which allow the straightforward synthesis of chiral sulfides, including β-hydroxysulfides bearing two adjacent C(sp³)−S and C(sp³)−O stereocentres, from simple and easily accessible materials, have also been achieved. Furthermore, the versatility of this method was demonstrated by the stereoconvergent synthesis of chiral compounds bearing stereocentres at C(sp³)−N/O/Se bonds. This methodology not only addresses the long-standing issues such as catalyst poisoning and C(sp²)/C(sp³)−S bond hydrogenolysis faced in TM-catalyzed hydrogenation of vinyl sulfides but also adds ENEs to the restricted repertoire of biocatalysts available for the construction of chiral C(sp³)−S bonds.

## Methods
### General

Column chromatography was carried out using Sigma Aldrich silica gel particle size, 40–63 μm particle size 60 Å. ¹H NMR, ¹³C NMR, ¹⁹F NMR, and NOESY were measured with Bruker Ascend400 Spectrometer, Bruker Avance III 400 or Bruker (Germany) Avance Neo 500 at room

temperature operating at the frequencies indicated. $^1$H NMR, $^{13}$C NMR, $^{19}$F NMR and NOESY analysis was carried out with MestReNova9.0. Mass spectra were recorded at the EPSRC National Mass Spectrometry Service Center on a Thermo Scientific LTQ Orbitrap XL Mass Spectrometer using low-resolution ESI or high-resolution nano ESI techniques. HPLC analysis was carried out using a Perkin-Elmer 1100 HPLC system coupled with UV/Vis set to the appropriate wavelength (214, 230, or 254 nm). The chiral columns used for HPLC analysis included Chiralpak® IG (4.6 mm × 250 mm, 5 μm), Chiralpak® ID column (4.6 mm × 250 mm, 5 μm), Chiralcel® OJ-H column (4.6 mm × 250 mm, 5 μm), Chiralcel® OD-H column (4.6 mm × 250 mm, 5 μm) and Chiralpak® IC column (4.6 mm × 250 mm, 5 μm) supplied by Daicel. Hexane, heptane, isopropanol (IPA), and ethanol (EtOH) were used as the eluent for all columns. $\alpha_D$ measurements were taken using a Bellingham and Stanley ADP440 + Polarimeter with a cell length of 0.5 dm. The homology model of ENE-101 enzymes was built from Glycerol Trinitrate Reductase NerA from *Agrobacterium radiobacter* (PDB ID: 4JIC, 66% sequence identity) using the Homology Modeling tool of the Schrodinger routine (version 2021-1). The Protein Preparation Wizard tool within the Schrodinger suite was applied to the structure of the enzyme to assign bond orders, add hydrogens, and perform a minimization with the OPLS3e force fields. The structures of the substrates were sketched with the 3D builder routine and submitted to LigPrep (Schrodinger suite) to generate input ligand structures for the next calculations. To set up docking simulations, the receptor grid generation routine of the Glide (grid-based ligand docking with energetics) software (version 9.0) was used to codify the shape and properties of the enzymatic binding pocket into a grid, in turn, used to score the ligand poses. The cofactor structure was used to define the grid, whose inner size was set to 14x14x14 Å to allow the substrate structure to find all the possible binding modes within the catalytic pocket. The hydroxyl groups of Ser, Thr, Tyr, and the thiol group of Cys were allowed to rotate and adopt different orientations for the most profitable interactions with different ligands. Extra-precision molecular docking simulations were performed with flexible ligand sampling and post-docking minimization of the resulting substrate/enzyme complexes. Fukui functions were calculated at the B3LYP-D3/LACVP* + level of theory by using the Jaguar software (version 11.1) and used to derive Fukui indices.

### General procedure for ENE-101 biocatalyzed enantioselective reduction of (Z)-1 and/or (E)-1 into (S)-2

To a solution of ENE-101 (25 mg, 65.8 U/mg, 1.2 mol‰), GDH-101 (12.5 mg, 32.0 U/mg) and D-glucose (0.5 mmol) in 1.7 mL KPBS (250 mM, pH 7.0) in a 10 mL vial, a solution of NADP$^+$ (5 μmol) in 200 μL KPBS (250 mM, pH 7.0) and a solution of (Z)-1 and/or (E)-1 (0.1 mmol) in DMSO (100 μL) were added. The vial was then incubated at 30 °C with shaking at 200 rpm for 24 h. After that, the reaction mixture was extracted with EtOAc (2 mL × 5) and centrifuged at 25 °C and 8000 rpm for 5 min, and the combined organic phase was dried with MgSO$_4$. An aliquot of the organic phase was taken for HPLC analysis to determine the conversions and ee values. Then the organic solvent was removed *in vacuo*, and the residue obtained was purified by column chromatography on silica gel to give the desired products 2.

### General procedure for two-step one-pot chemoenzymatic cascade for the enantioselective synthesis of (S)-2

3.6 mL KPBS (250 mM, pH 7.0) was added to a solution of 3 (0.1 mmol) and 4a (0.1 mmol) in DMSO (200 μL) in a 10 mL vial, and the resulting mixture was stirred at 30 °C for 24 h. After that, ENE-101 (50 mg, 65.8 U/mg, 2.5 mol‰), GDH-101 (25 mg, 32.0 U/mg), D-glucose (1 mmol), and a solution of NADP$^+$ (10 μmol) in 200 μL KPBS (250 mM, pH 7.0) were added sequentially. After addition, the vial was incubated at 30 °C with shaking at 200 rpm for 24 h. Then the reaction mixture was

extracted with EtOAc (2 mL × 5) and centrifuged at 25 °C and 8000 rpm for 5 min, and the combined organic phase was dried with MgSO$_4$. An aliquot of the organic phase was taken for HPLC analysis to determine the conversions and ee values. Then the organic solvent was removed *in vacuo*, and the residue obtained was purified by column chromatography on silica gel to give the desired products 2.

### General procedure for one-step one-pot chemoenzymatic cascade for the enantioselective synthesis of (S)-2

To a solution of 3 (0.1 mmol) and 4a (0.1 mmol) in DMSO (200 μL) in a 10 mL vial, 3.6 mL KPBS (250 mM, pH 7.0), ENE-101 (50 mg, 65.8 U/mg, 2.5 mol‰), GDH-101 (25 mg, 32.0 U/mg), D-glucose (1 mmol), and a solution of NADP$^+$ (10 μmol) in 200 μL KPBS (250 mM, pH 7.0) were added. The vial was then incubated at 30 °C with shaking at 200 rpm for 24 h. Then the reaction mixture was extracted with EtOAc (2 mL × 5) and centrifuged at 25 °C and 8000 rpm for 5 min, and the combined organic phase was dried with MgSO$_4$. An aliquot of the organic phase was taken for HPLC analysis to determine the conversions and ee values. Then the organic solvent was removed *in vacuo*, and the residue obtained was purified by column chromatography on silica gel to give the desired products 2.

### General procedure for ENE-101 biocatalyzed enantioselective reduction of 5 or 6 into (S)-7 or 8

To a solution of ENE-101 (25 mg, 65.8 U/mg, 1.2 mol‰), GDH-101 (12.5 mg, 32.0 U/mg), and D-glucose (0.5 mmol) in 1.7 mL KPBS (250 mM, pH 7.0) in a 10 mL vial, a solution of NADP$^+$ (5 μmol) in 200 μL KPBS (250 mM, pH 7.0) and a solution of 5 or 6 (0.1 mmol) in DMSO (100 μL) were added. The vial was then incubated at 30 °C with shaking at 200 rpm for 1–24 h. After that, the reaction mixture was extracted with EtOAc (2 mL × 5) and centrifuged at 25 °C and 8000 rpm for 5 min, and the combined organic phase was dried with MgSO$_4$. An aliquot of the organic phase was taken for HPLC analysis to determine the conversions and ee values. Then the organic solvent was removed *in vacuo*, and the residue obtained was purified by column chromatography on silica gel to give the desired products 7 or 8.

### General procedure for one-step one-pot hydrogen-borrowing cascade for the enantioselective synthesis of (S)-7 from 9

To a solution of ENE-101 (5 mg, 65.8 U/mg, 1.2 mol‰) and ADH-159 (5 mg, 0.2 U/mg, 1.9 mol‰) in 0.85 mL KPBS (250 mM, pH 7.0) in a 2 mL vial, a solution of NADP$^+$ (1 μmol) in 100 μL KPBS (250 mM, pH 7.0) and a solution of 9 (0.02 mmol) in DMSO (50 μL) were added, and the vial was then incubated at 30 °C with shaking at 200 rpm for 1.5 or 12 h. After that, the reaction mixture was extracted with EtOAc (0.5 mL × 5) and centrifuged at 25 °C and 8000 rpm for 5 min, and the combined organic phase was dried with MgSO$_4$. An aliquot of the organic phase was taken for HPLC analysis to determine the conversions and ee values.

### General procedure for two-step one-pot biocatalytic-chemical (bio-chem) or biocatalytic-biocatalytic (bio-bio) cascade for the enantioselective synthesis of 10 from 5

To a solution of ENE-101 (25 mg, 65.8 U/mg, 1.2 mol‰), GDH-101 (12.5 mg, 32.0 U/mg), and D-glucose (0.5 mmol) in 1.7 mL KPBS (250 mM, pH 7.0) in a 10 mL vial, a solution of NADP$^+$ (5 μmol) in 200 μL KPBS (250 mM, pH 7.0) and a solution of 5 (0.1 mmol) in DMSO (100 μL) were added, and the vial was then incubated at 30 °C with shaking at 200 rpm for 1.5, 2 or 3 h. Subsequently, the reaction was added with either a) MeOH (1 mL) and NaBH$_4$ (0.5 mmol) with stirring at room temperature for 1 h, or b) ADH-153 (50 mg, 3.3 U/mg, 3.8 mol‰) or ADH-19 (50 mg, 2.8 U/mg, 3.8 mol‰) with incubation at 30 °C and shaking at 200 rpm for 18 h. After that, the reaction mixture was extracted with EtOAc (2 mL × 5) and centrifuged at 25 °C and 8000 rpm

for 5 min, and the combined organic phase was dried with MgSO$_4$. An aliquot of the organic phase was taken for HPLC analysis to determine the conversions, dr, and ee values. Then the organic solvent was removed *in vacuo*, and the residue obtained was purified by column chromatography on silica gel to give the desired products **10**.

## General procedure for one-step one-pot biocatalytic-biocatalytic (bio-bio) cascade for the enantioselective synthesis of 10 from 5

To a solution of ENE-101 (25 mg, 65.8 U/mg, 1.2 mol‰), ADH-153 (50 mg, 3.3 U/mg, 3.8 mol‰) or ADH-19 (50 mg, 2.8 U/mg, 3.8 mol‰), GDH-101 (12.5 mg, 32.0 U/mg) and D-glucose (0.5 mmol) in 1.7 mL KPBS (250 mM, pH 7.0) in a 10 mL vial, a solution of NADP$^+$ (5 μmol) in 200 μL KPBS (250 mM, pH 7.0) and a solution of **5** (0.1 mmol) in DMSO (100 μL) were added, and the vial was then incubated at 30 °C with shaking at 200 rpm for 18 h. After that, the reaction mixture was extracted with EtOAc (2 mL × 5) and centrifuged at 25 °C and 8000 rpm for 5 min, and the combined organic phase was dried with MgSO$_4$. An aliquot of the organic phase was taken for HPLC analysis to determine the conversions, dr, and ee values. Then the organic solvent was removed *in vacuo*, and the residue obtained was purified by column chromatography on silica gel to give the desired products **10**.

## General procedure for three-step one-pot biocatalytic-biocatalytic-biocatalytic (bio-bio-bio) cascade for the enantioselective synthesis of 10 from 9

To a solution of ADH-159 (25 mg, 0.2 U/mg, 1.9 mol‰) or ADH-153 (25 mg, 3.3 U/mg, 1.9 mol‰) or ADH-19 (25 mg, 2.8 U/mg, 1.9 mol‰) and NOX-009 (25 mg) in 3.6 mL KPBS (250 mM, pH 7.0) in a 10 mL vial, a solution of NADP$^+$ (5 μmol) in 200 μL KPBS (250 mM, pH 7.0) and a solution of **9** (0.1 mmol) in DMSO (200 μL) were added, and the vial was then incubated at 30 °C with shaking at 200 rpm for 24 h. Then ENE-101 (25 mg, 65.8 U/mg, 1.2 mol‰), GDH-101 (12.5 mg, 32.0 U/mg), and D-glucose (0.5 mmol) were added sequentially, and the vial was incubated at 30 °C with shaking at 200 rpm for 8 h. Then the reaction mixture was extracted with EtOAc (2 mL × 5) and centrifuged at 25 °C and 8000 rpm for 5 min, and the combined organic phase was dried with MgSO$_4$. An aliquot of the organic phase was taken for HPLC analysis to determine the conversions and ee values. Then the organic solvent was removed *in vacuo*, and the residue obtained was purified by column chromatography on silica gel to give the desired products **10**.

## Reporting summary

Further information on research design is available in the Nature Portfolio Reporting Summary linked to this article.

## Data availability

All relevant data supporting the findings of this study, including experimental procedures, characterization of substrates and products, computational details, copies of $^1$H NMR, $^{13}$C NMR, $^{19}$F NMR, and NOESY spectra, and copies of HPLC spectra are available within the paper and its Supplementary Information. All data supporting the findings of this study are also available from the corresponding authors upon request.

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

## Acknowledgements

We gratefully acknowledge BBSRC (LIDo iCASE Studentship BB/T008709/1 to AM) for financial support. For F.Z. and D.C., this project has received funding from the European Union's Horizon 2020 research and innovation program under the Marie Skłodowska-Curie grant agreement 838326.

## Author contributions

D.C. and B.D. conceived and directed the project. F.Z., A.M., and R.A. performed the experimental studies. F.M. and A.P. carried out the computational studies. S.M. contributed to the discussion and decision making of the project and to the supervision of A.M. All the authors contributed to the writing of the manuscript.

## Competing interests

The authors declare no competing interests.
