## [Peer Review File · Nature Communications]

Cooperative chemoenzymatic and biocatalytic cascades to access chiral sulfur compounds bearing C(sp³)-S stereocentresREVIEWER COMMENTS

Reviewer #1 (Remarks to the Author):

In this paper, the authors have used ene reductase in particular ENE-101 for the stereoselective reduction of alkene bond of vinyl sulfides to obtain chiral sulfur compounds with excellent enantioselectivities and high yields. Previously such compounds are prepared using transition metal catalysts complexes together with chiral phosphine ligands, which may cause the poisoning of the metal catalysts used. Hence, the authors developed a stereoconvergent reduction catalysed by ene reductase where both (E)- as well as (Z)-vinyl sulfide substrates gave (S)-configured sulfides. The same method was further improved by designing a two step one-pot chemoenzymatic cascade where the thiophenol and DMAD were used as a starting material in KPBS buffer (pH 7.0) to get chiral sulfides. This allowed the direct synthesis of enantiopure sulfides from commercially available material in just one step.

The mechanistic studies were also performed to investigate the reason for the origin of stereoconvergent nature of the ene reductase catalysed reductions. In addition, deuterium labelling experiments were performed to identify the site of hydride attack on the carbon-carbon double bond of (Z)-1aa. To further diversify the application of the ene biocatalyst, the authors have first shown the reduction of vinyl sulfides containing only one electron withdrawing group (mainly methyl ketone), and then extended to the synthesis of beta-hydroxysulfides which are obtained in excellent ee and dr using alcohol dehydrogenase (ADH) and ENE-101 cascade in one pot.

Although, the biocatalytic method gave access to number of substituted chiral sulfides and hydroxysulfides, all the enzymes used in the study are purchased from commercial sources. As per the results shown in table 1, certainly, there is good chance that already known ene reductases such as OYE, Nostoc ER or NtDBR etc may also give similar results in terms of yields and ee. The authors should have explored the possibility to identify ene reductases which could give access to (R)-configured sulfides. This would have made the study more interesting. In addition, some products have been obtained first using the vinyl sulfides and then making them directly from commercially available starting material. Authors may have done optimization with few substrates and then shown one step chemoenzymatic synthesis of chiral sulfides as shown in Figure 3a-b.

Similar strategy was used in the synthesis of hydroxysulfides where first the two enzymes ADH and ENE-102 were used in two steps and then in one pot to arrive at the same molecules.

Overall, it is a nice work but the results could have been presented in a more concise manner without repeating the results and new enzymes might have been used in the study.

Comments:

1. In the first paragraph after figure 2, authors wrote excellent enantioselectivities (79->99% ee), and good to excellent conversions (83->99%). The ee can be changed to "good to excellent"

2. In chemoenzymatic cascade synthesis of sulfides 2. (3rd line), the authors wrote Delightfully, thiophenol reacted smoothly with DMAD in a KPBS (pH 7.0)/DMSO (18/1) solution at 30 oC for 24 h, leading to its full conversion into the corresponding isomeric mixture of vinyl sulfides (Z)-1aa and (E)-1aa, which were then reduced by ENE-101 added in one-pot to give (S)-2aa in stereoconvergent manner, good overall yield (65%) and excellent enantioselectivity (92% ee).

It is suggested that excellent can be changed to good ee

3. The authors wrote "Noteworthy, no loss in enantioselectivity was observed as compared with the direct reduction of the corresponding vinyl sulfides/selenide, suggesting the efficiency of the two-step one-pot cascade."

The above sentence needs some modification since in figure 2, compound 2aa, 2ab and 2ac and so on the ee shown is >99% in most cases, however in figure 3, the ee for these compounds gets lowered.

4. No information about the docking procedure used and the sequence that was used to make homology model of ENE-101 was given.

Reviewer #2 (Remarks to the Author):

In this manuscript Zhao et al report about the biocatalytic synthesis of chiral thioether derivatives by asymmetric reduction of prochiral thioenoether substrates using ene reductases in good yields and good to excellent enantioselectivities. As the authors correctly argue in their introduction the synthesis of chiral thio containing compounds represents a considerable challenge for transition metal catalysis as catalyst poisoning is an often

observed problem. In contrast, biocatalytic methods have mostly focused on kinetic resolutions of racemic mixtures.

The authors present a very attractive solution to this synthetic challenge by the biocatalytic reduction of readily accessible vinylthioethers using an ene reductase (“ENE-101”) using either a NAD(P)H cofactor regeneration system or by combination with other enzymes (ADH) using a hydrogen borrowing mechanism.

The scope and limitations of the transformations has been studied in great detail (Figures 2-5) and document the impressive tolerance to various substituents at the thio substituent and also on the alkene. While many substrates bear two EWG, in Figure 4 also substrates with a ketone as sole EWG have been successfully converted.

In Figure 3c and 3d deuteration experiments and docking experiments are depicted which suggest a hydride transfer to the less substituted olefin carbon, followed by protonation of the resulting enolate at the higher substituted olefin carbon.

I was very impressed by this work and see it as a great manuscript for Nat. Commun. due to its originality and novelty, its practical implications and its extensive experimental documentation.

Before publication the following issues have to be addressed:

1) At its present state, it will be not possible to reproduce this work, as the authors do not provide sufficient information about the used enzymes. In the SI they write on page 4 “All the enzymes including ENEs, GDHs, ADHs used in this study are cell-free extracts, which are available through Johnson Matthey for the replication of this work.¹” and state Ref. 1, which refers to a patent of Johnson Matthey. But how could I get access to the enzyme? Can I purchase it from Johnson Matthey (what is the article number?), or can I get it for free (Whom should I contact? What is the email address? What are the requirements? Do I have to sign an MTA?)

It is good practice in biocatalysis research to publish the gene sequence of the enzyme. This should be included in the SI.

I understand that Johnson Matthey might have some economic interest in this research, but this should not compromise the scientific content and reproducibility of a publication.

2) As “ENE-101” is none of the standard enzymes, it was not clear to me, if the authors worked with a purified enzyme or cell lysate, when reading the manuscript. The authors should write already when discussing the screening in Table 1 that they used cell free extracts.

The authors should also state the molar concentration of the ene reductase catalyst and present the TON of representative reactions.

3) The authors sometimes use hyperboles for the description of enantioselectivities. While there is no precise definition of the verbal description of enantioselectivities, I believe that most chemists would agree that 88% ee is neither “high enantioselectivity” nor “excellent selectivities” as phrased by the authors on page 3 or 72% ee as “excellent selectivities” on page 4.

The authors should use a consistent language and might declare ee’s below 90% as “good” and around 50% as “moderate” and reserve “excellent” for >98%

4) The authors do not use a consistent style to write the stereochemical descriptors of compound names in bold or normal font. Many times, but not always, (E/Z) are written bold, and (R/S) almost never.

5) The authors should include the following reference in their introduction:

Synthesis of Enantiopure Sulfoxides by Concurrent Photocatalytic Oxidation and Biocatalytic Reduction

S. Bierbaumer, L. Schermund, A. List, C. K. Winkler, S. M. Glueck, W. Kroutil

Angew. Chem. Int. Ed. 2022, e202117103

6) In Fig. 4 the authors claim that compound 7af was produced in 93% ee. However, when looking at the HPLC traces on page 278 of the SI the retention times of the chiral HPLC measurement of the enzymatic reaction (third chromatogram) has no correlation with the ones of the racemic mixture (second chromatogram). Therefore, it is even not clear if the compound has been produced at all, and if the HPLC trace are the ones of the discussed reaction!!!

The authors should check their data for all chromatograms and ideally present the same section of the HPLC trace for both racemic standard as well as reaction mixture, so that the reader can immediately see if the peaks elute at the same retention times. [The authors have done this for many but not all compounds.]

In conclusion, I can recommend the publication of this very interesting manuscript in Nat. Commun. once the issues raised in this report will have been addressed.

Minor comments:

Fig.1. sp³, sp² etc should be written with superscripted numbers

Fig. 3d. Z and E should be written in Italics.

Reviewer #3 (Remarks to the Author):

The manuscript “Cooperative chemoenzymatic and biocatalytic cascades for the enantioselective synthesis of chiral sulfur compounds bearing stereocentres at C(sp³)-S bond” describes a challenging combination of new processes that all in themselves are not revolutionary. Devil is thus in the detail.

What are the noteworthy results?

The core results are in table 1 exploring the scope of the ENE, also known as ERED and the recycling enzyme – not very exciting GDH. Subsequently this is combined with and ADH in a nice manner to keep the reducing equivalents in Fig. 4. This is similar to Chem. Commun.,

2012, 48, 6630–6632. for fig. 4b the authors should also consider the influence of thermodynamics on the yield. The stability of 7 over 9 is the driving force and if the differences between 9 and 7 are not large it will essentially stay in equilibrium.

The real highlight is then in fig. 5. The set up of two stereocenters by two sequential enzymes is very nice.

Will the work be of significance to the field and related fields? How does it compare to the established literature? If the work is not original, please provide relevant references.

The work that is the basis for fig. 5 is relevant and new. It would be nice if it could be expanded also with an ADH with the opposite stereoselectivity.

Does the work support the conclusions and claims, or is additional evidence needed?

In addition to conversion after 24 h a time course experiment would add significant information.

Are there any flaws in the data analysis, interpretation and conclusions? Do these prohibit publication or require revision?

There are no flaws. But it would be nice to receive information on the stability of 7aj.

Is the methodology sound? Does the work meet the expected standards in your field?

The methodology is sound but it also falls short. All enzymes are used as cell free extract in mg and no activity assay is provided. This means that the decision to take one enzyme is always based on the coincidental activity in the cell free extract with might fluctuate considerably between batches. To correctly describe enzymatic reactions see Gardossi, L., Poulsen, P. B., Ballesteros, A., Hult, K., Svedas, V. K., Vasic-Racki, D. et al. (2010) Guidelines for reporting of biocatalytic reactions Trends in Biotechnology 28, 171-180
10.1016/j.tibtech.2010.01.001.

Is there enough detail provided in the methods for the work to be reproduced?

See comment above.

Response to Reviewers' Comments

Reviewer #1 (Remarks to the Author):

Comment: In this paper, the authors have used ene reductase in particular ENE-101 for the stereoselective reduction of alkene bond of vinyl sulfides to obtain chiral sulfur compounds with excellent enantioselectivities and high yields. Previously such compounds are prepared using transition metal catalysts complexes together with chiral phosphine ligands, which may cause the poisoning of the metal catalysts used. Hence, the authors developed a stereoconvergent reduction catalysed by ene reductase where both (*E*)- as well as (*Z*)-vinyl sulfide substrates gave (*S*)-configured sulfides. The same method was further improved by designing a two-step one-pot chemoenzymatic cascade where the thiophenol and DMAD were used as a starting material in KPBS buffer (pH 7.0) to get chiral sulfides. This allowed the direct synthesis of enantiopure sulfides from commercially available material in just one step. The mechanistic studies were also performed to investigate the reason for the origin of stereoconvergent nature of the ene reductase catalysed reductions. In addition, deuterium labelling experiments were performed to identify the site of hydride attack on the carbon-carbon double bond of (*Z*)-1aa. To further diversify the application of the ene biocatalyst, the authors have first shown the reduction of vinyl sulfides containing only one electron withdrawing group (mainly methyl ketone), and then extended to the synthesis of beta-hydroxysulfides which are obtained in excellent ee and dr using alcohol dehydrogenase (ADH) and ENE-101 cascade in one pot. Although, the biocatalytic method gave access to number of substituted chiral sulfides and hydroxysulfides, all the enzymes used in the study are purchased from commercial sources. As per the results shown in table 1, certainly, there is good chance that already known ene reductases such as OYE, Nostoc ER or NtDBR etc may also give similar results in terms of yields and ee. The authors should have explored the possibility to identify ene reductases which could give access to (*R*)-configured sulfides. This would have made the study more interesting. In addition, some products have been obtained first using the vinyl sulfides and then making them directly from commercially available starting material. Authors may have done optimization with few substrates and then shown one step chemoenzymatic synthesis of chiral sulfides as shown in Figure 3a-b. Similar strategy was used in the synthesis of hydroxysulfides where first the two enzymes ADH and ENE-101 were used in two steps and then in one pot to arrive at the same molecules. Overall, it is a nice work but the results could have been presented in a more concise manner without repeating the results and new enzymes might have been used in the study.

Response: Thanks for your positive comments. As an industry & academia collaboration project, we aimed to address some synthetic challenges (namely the construction of chiral C(sp³)-S bond, a major challenge in organic synthetic chemistry) with the enzyme products developed by the industrial co-authors (Johnson Matthey). We were pleased to find that the commercially available enzymes, and in particular ENE-101, from Johnson Matthey worked well on the substrates of interest without the need of further enzyme evolution. To address the reviewer's comments, we have performed and added the following experiments into our revised manuscript.

(a) Three other ENEs (namely NCR, pQR1440 and pQR1907), available in our laboratory at UCL have been tested on substrates (*Z*)-1aa, (*E*)-1aa and (*Z*)-5aa. None of them provided the opposite (*R*)-configured sulfides while instead they afforded the (*S*)-configured sulfides with moderate to high ee values. However, ENE-105 did deliver the opposite (*R*)-7aa, albeit with only 17% ee. Detailed information is now reported in Supplementary Table 3 and 9 in our revised Supplementary

Information.

(b) We did optimize the reaction condition for the chemoenzymatic cascades for the synthesis of (S)-2aa before exploring the substrate scope shown in Figure 3a-b. Reaction parameters such as molecular ratio of 3a/4a, the amount of ENE-101/GDH-101/D-glucose/NADP⁺, reaction time and reaction temperature were optimized. The optimization results can be found in Supplementary Table 5 and 7 in our revised Supplementary Information.

Comment: 1. In the first paragraph after figure 2, authors wrote excellent enantioselectivities (79->99% ee), and good to excellent conversions (83->99%). The ee can be changed to “good to excellent”

Response: Changed. Thank you for pointing it out.

Comment: 2. In chemoenzymatic cascade synthesis of sulfides 2. (3rd line), the authors wrote Delightfully, thiophenol reacted smoothly with DMAD in a KPBS (pH 7.0)/DMSO (18/1) solution at 30 oC for 24 h, leading to its full conversion into the corresponding isomeric mixture of vinyl sulfides (Z)-1aa and (E)-1aa, which were then reduced by ENE-101 added in one-pot to give (S)-2aa in stereoconvergent manner, good overall yield (65%) and excellent enantioselectivity (92% ee). It is suggested that excellent can be changed to good ee

Response: Changed. Thank you for pointing it out.

Comment: 3. The authors wrote “Noteworthy, no loss in enantioselectivity was observed as compared with the direct reduction of the corresponding vinyl sulfides/selenide, suggesting the efficiency of the two-step one-pot cascade.” The above sentence needs some modification since in figure 2, compound 2aa, 2ab and 2ac and so on the ee shown is >99% in most cases, however in figure 3, the ee for these compounds gets lowered.

Response: Thanks for your suggestion, and we agree that this statement could be misleading. We have changed it into “no obvious loss in enantioselectivity was observed...” in our revised manuscript.

Comment: 4. No information about the docking procedure used and the sequence that was used to make homology model of ENE-101 was given.

Response: The sequence of ENE-101 is now provided in the Supporting Information (section II, Biology). The sequence identity (66%) between ENE-101 and Glycerol Trinitrate Reductase NerA from *Agrobacterium radiobacter* (PDB ID: 4JIC) used for the construction of the homology model is also stated in the Supporting Information (section IV).

Reviewer #2 (Remarks to the Author):

Comment: In this manuscript Zhao et al report about the biocatalytic synthesis of chiral thioether derivatives by asymmetric reduction of prochiral thioenolether substrates using ene reductases in good yields and good to excellent enantioselectivities. As the authors correctly argue in their introduction the synthesis of chiral thio containing compounds represents a considerable challenge for transition metal catalysis as catalyst poisoning is an often observed problem. In contrast, biocatalytic methods have mostly focused on kinetic resolutions of racemic mixtures. The authors

present a very attractive solution to this synthetic challenge by the biocatalytic reduction of readily accessible vinylthioethers using an ene reductase (“ENE-101”) using either a NAD(P)H cofactor regeneration system or by combination with other enzymes (ADH) using a hydrogen borrowing mechanism. The scope and limitations of the transformations has been studied in great detail (Figures 2-5) and document the impressive tolerance to various substituents at the thio substituent and also on the alkene. While many substrates bear two EWG, in Figure 4 also substrates with a ketone as sole EWG have been successfully converted. In Figure 3c and 3d deuteration experiments and docking experiments are depicted which suggest a hydride transfer to the less substituted olefin carbon, followed by protonation of the resulting enolate at the higher substituted olefin carbon. I was very impressed by this work and see it as a great manuscript for Nat. Commun. due to its originality and novelty, its practical implications and its extensive experimental documentation.

Response: Thank you for your positive feedback as well as recognition of the importance of our work.

Comment: Before publication the following issues have to be addressed:

1) At its present state, it will be not possible to reproduce this work, as the authors do not provide sufficient information about the used enzymes. In the SI they write on page 4 “All the enzymes including ENEs, GDHs, ADHs used in this study are cell-free extracts, which are available through Johnson Matthey for the replication of this work.¹” and state Ref. 1, which refers to a patent of Johnson Matthey. But how could I get access to the enzyme? Can I purchase it from Johnson Matthey (what is the article number?), or can I get it for free (Whom should I contact? What is the email address? What are the requirements? Do I have to sign an MTA?) It is good practice in biocatalysis research to publish the gene sequence of the enzyme. This should be included in the SI. I understand that Johnson Matthey might have some economic interest in this research, but this should not compromise the scientific content and reproducibility of a publication.

Response: All the enzymes, including ENEs, ADHs and GDHs used in this study, are commercially available from Johnson Matthey via <https://matthey.com> for the replication of this work. The product code for the ENE kit is EZK002, which includes 7 ENEs, 3 GDHs, 1 FDH, NAD⁺ and NADP⁺. The product code for the ADH kit is EZK003, which includes 17 ADHs, 3 GDHs, 1 FDH, NAD⁺ and NADP⁺. All the enzymes are also available from Johnson Matthey for cooperative research purposes by contacting Dr Beatriz Dominguez (beatriz.dominguez@matthey.com), and a material transfer agreement (MTA) needs to be signed in this case.

The gene and amino acid sequence of ENE-101 have been added into our revised Supplementary Information as they are already patented by Johnson Matthey (please see reference 1 in the Supporting Information). Since ADH-19, ADH-153 and ADH-159 are protected by the commercial intellectual property (IP) of Johnson Matthey, their gene sequences are not publicly available. This kind of situation has been analyzed and discussed in the literature (please see a viewpoint: the section of “The biocatalysis industry model: why disclosing detailed enzyme information is tricky business” in *ACS Med. Chem. Lett.* 2019, 10, 1363–1366.), but this doesn’t prevent people from adopting enzymes for organic synthesis purpose. As a matter of fact, commercial enzymes with unknown gene sequences have been widely used in pharmaceutical industry for the manufacturing of active pharmaceutical ingredients and key intermediates thereof (please see a review: The Evolving Nature of Biocatalysis in Pharmaceutical Research and Development. *JACS Au* 2023, 3,

715–735). Therefore, we believe that it is acceptable to refer enzymes to the commercial source without gene sequences for publications in biocatalysis.

The abovementioned information has been added into the revised Supplementary Information. Besides, the enzyme activity assays of ENEs/GDHs/ADHs (Supplementary Table 1 and 2) have also been added into our revised Supplementary Information.

Comment: 2) As “ENE-101” is none of the standard enzymes, it was not clear to me, if the authors worked with a purified enzyme or cell lysate, when reading the manuscript. The authors should write already when discussing the screening in Table 1 that they used cell free extracts. The authors should also state the molar concentration of the ene reductase catalyst and present the TON of representative reactions.

Response: Thanks for your comment. According to your suggestion, now we indicate that we used cell-free extracts when discussing the screening in Table 1.

We also indicate the molar catalyst loading (molar ratios in mol%) of ENEs and ADHs in all reactions. The TON of representative reactions (products 2aa and 7aa) is also reported in the manuscript.

Comment: 3) The authors sometimes use hyperboles for the description of enantioselectivities. While there is no precise definition of the verbal description of enantioselectivities, I believe that most chemists would agree that 88% ee is neither “high enantioselectivity” nor “excellent selectivities” as phrased by the authors on page 3 or 72% ee as “excellent selectivities” on page 4. The authors should use a consistent language and might declare ee’s below 90% as “good” and around 50% as “moderate” and reserve “excellent” for >98%

Response: Thanks for your helpful suggestion, and we have made the corresponding changes throughout the manuscript.

Comment: 4) The authors do not use a consistent style to write the stereochemical descriptors of compound names in bold or normal font. Many times, but not always, (E/Z) are written bold, and (R/S) almost never.

Response: The stereochemical descriptors such as *E/Z* and *R/S* are now written in normal font without bold throughout the revised manuscript and Supplementary Information. However, they are still written in bold if they are located in a subheading as requested by Formatting Instructions of Nature Communications.

Comment: 5) The authors should include the following reference in their introduction:
Synthesis of Enantiopure Sulfoxides by Concurrent Photocatalytic Oxidation and Biocatalytic Reduction S. Bierbaumer, L. Schmermund, A. List, C. K. Winkler, S. M. Glueck, W. Kroutil *Angew. Chem. Int. Ed.* 2022, e202117103

Response: Thanks for your comment. The above reference has been cited as Ref. 31 in our revised manuscript.

Comment: 6) In Fig. 4 the authors claim that compound **7af** was produced in 93% ee. However, when looking at the HPLC traces on page 278 of the SI the retention times of the chiral HPLC

measurement of the enzymatic reaction (third chromatogram) has no correlation with the ones of the racemic mixture (second chromatogram). Therefore, it is even not clear if the compound has been produced at all, and if the HPLC trace are the ones of the discussed reaction!!! The authors should check their data for all chromatograms and ideally present the same section of the HPLC trace for both racemic standard as well as reaction mixture, so that the reader can immediately see if the peaks elute at the same retention times. [The authors have done this for many but not all compounds.] In conclusion, I can recommend the publication of this very interesting manuscript in Nat. Commun. once the issues raised in this report will have been addressed.

Response: Thanks for your helpful suggestion, we agree that the result of **7af** was misleading. We have repeated this reaction and obtained almost same data of conversion (88%), yield (72%) and ee value (94%). We have updated the data in our revised manuscript and HPLC spectra in the revised Supplementary Information. The correlation of retention time of substrate and product is good and reliable now.

Besides, according to your suggestion, we have checked all the HPLC spectra, and updated the HPLC spectra by zooming in/out the original spectra (in the case of **2ae** (page 231), **2av** (page 248), **7aa** (page 297), **10ad** (page 310)) or repeating the corresponding experiments (in the case of **2ak** (page 237, 266, 278)) to make sure the same section of the HPLC traces for substrates, racemic standards as well as reaction mixture is presented.

Comment: Minor comments: Fig.1. sp3, sp2 etc should be written with superscripted numbers. Fig. 3d. Z and E should be written in Italics.

Response: Changed. Thank you for pointing it out.

Reviewer #3 (Remarks to the Author):

Comment: The manuscript “Cooperative chemoenzymatic and biocatalytic cascades for the enantioselective synthesis of chiral sulfur compounds bearing stereocentres at C(sp³)-S bond” describes a challenging combination of new processes that all in themselves are not revolutionary. Devil is thus in the detail.

General Response: Thanks for your comment. We believe that this work is novel and challenging from a synthetic chemistry point of view, as also highlighted and recognized by the other two reviewers.

(a) As stated in the introduction, the transition-metal-catalysed asymmetric hydrogenation of vinyl sulfides is the best synthetic method to access enantiomerically pure sulfides from prochiral precursors. However, the few existing synthetic methods suffer from many disadvantages and scientific issues such as the need of poorly sustainable transition metals, complex chiral phosphine ligands and, in some cases, harsh conditions (*e.g.* high pressure, high temperature), as well as the catalyst poisoning by the strongly coordinating divalent sulfur atom and the C-S bond hydrolysis. This clearly shows that the asymmetric reduction of vinyl sulfides is a major synthetic challenge faced in modern organic chemistry. Herein, we adopted biocatalysis to circumvent the limitations of traditional chemistry. We developed a ENE-101 catalysed asymmetric reduction of vinyl sulfides, which successfully address these scientific issues and allowed the efficient preparation of enantiomerically pure sulfides bearing stereocentres at C-S bond under mild and more sustainable conditions. This work is a good example of showing how enzymes can offer

better solutions to synthetic challenges than traditional chemical methods.

(b) Although biocatalysis has been widely used for the construction of carbon-carbon/heteroatom stereocentres, yet its application in chiral C(sp³)-S bond construction is rare and limited to enzymatic kinetic resolution reactions. To the best of our knowledge, there is no report of construction of chiral C(sp³)-S bond from prochiral precursors via biocatalytic asymmetric catalysis.

(c) A series of one-pot cooperative sequential/concurrent chemoenzymatic and biocatalytic cascades have also been developed for the facile assembly of chiral sulfides bearing C(sp³)-S stereocentres. Enzymatic cascades offer the possibility to access target compounds from easily available substrates and avoid the purification of reaction intermediates, thus improving the green and sustainability metrics of the process. In addition, this methodology has also been extended to the construction of chiral C(sp³)-N/O/Se bonds as well as valuable β -hydroxysulfides bearing two adjacent C(sp³)-S and C(sp³)-O stereocentres, further highlighting the application and potential of this work both in academia and industry.

We have carefully revised and improved the manuscript according to your helpful comments as well as the comments from other two reviewers, by adding the protein sequence of the enzymes, activity assays of enzymes, detailed experimental procedures for all reactions, reaction optimization of chemoenzymatic cascades shown in Fig. 3 and updated HPLC traces.

Comment: The core results are in table 1 exploring the scope of the ENE, also known as ERED and the recycling enzyme – not very exciting GDH. Subsequently this is combined with an ADH in a nice manner to keep the reducing equivalents in Fig. 4. This is similar to Chem. Commun., 2012, 48, 6630–6632. For fig. 4b the authors should also consider the influence of thermodynamics on the yield. The stability of 7 over 9 is the driving force and if the differences between 9 and 7 are not large it will essentially stay in equilibrium.

Response: This project developed an efficient methodology for the synthesis of enantiomerically pure sulfides employing ENE reductases developed by Johnson Matthey. The GDH regeneration system is a standard in ENE biocatalysis. GDH regeneration system is robust, economic and widely used in industry and academia, and it worked really well to help us achieve the main goals of this synthetic chemistry project.

The hydrogen-borrowing cascade (HBC) shown in Fig. 4 is only a minor part of our current work, and it was clearly inspired by previous literature (i.e. Chem. Commun. 2012, 48, 6630–6632). Following reviewer's suggestion and the procedure reported by Hollmann and co-workers (Chem. Commun. 2012, 48, 6630–6632, now cited as ref. 52 in our revised manuscript), we have considered the influence of thermodynamics on the formation of compound 7 starting from compound 9. In particular, a favorable thermodynamic driving force for the transformation of 9 into 7 is expected in the case a favorable difference in the Gibbs energy of formation of starting material and product is found, as also exemplified in the work reported by Hollmann. The group-contribution method (with corrections proposed by Mavrovouniotis M. L. in *Biotechnol. Bioeng.* 1990, 36, 1070-1082) was applied for the estimation of the Gibbs energy of formation of both 9 and 7. Briefly, the syntactic formula of each compound is broken down into chemical groups; each group contributes with a

fixed value to calculate the final Gibbs energy of formation of starting material and product; finally, the difference between Gibbs energy of formation of product minus Gibbs energy of formation of starting material will result in the Gibbs energy of reaction. As a result, Gibbs energies of formation calculated for **7** and **9** were 56.6 and 66.2 kcal/mol, respectively, which led to an overall balance of -9.6 kcal/mol. Similar values were obtained by application of additional corrections, such as those for conjugation, as reported by Hatzimanikatis and co-workers (*Biophys. J.* **2008**, *95*, 1487-1499). In this case, a favorable Gibbs energy of reaction (-13.8 kcal/mol) was found. These results suggest that conversion of **9** into **7** does occur. A sentence with related references has been added to the main manuscript.

Comment: The real highlight is then in fig. 5. The set up of two stereocenters by two sequential enzymes is very nice.

Response: Thanks for the positive comment. The one-pot biocatalytic cascades are indeed one of the highlights of this work, showing how different classes of enzymes can work cooperatively in a cascade manner to construct more complex and challenging molecules in a stereodefined manner.

Comment: Will the work be of significance to the field and related fields? How does it compare to the established literature? If the work is not original, please provide relevant references.

The work that is the basis for fig. 5 is relevant and new. It would be nice if it could be expanded also with an ADH with the opposite stereoselectivity.

Response: Thank you for your constructive suggestion. According to your suggestion, we carried out a series of experiments with ADH-19 to give the opposite *anti*-diastereoisomers as the major products with good to excellent enantioselectivities. Specifically:

- (a) the two-step one-pot ENE-101/ADH-19 cascade overall gave the opposite *anti*-diastereoisomers (*2R,3S*)-**10aa-ae** (5 examples) as the major products with 95->99% ee values (now in Fig. 6a, data shown in green);
- (b) the one-step one-pot ENE-101/ADH-19 cascade gave the opposite *anti*-diastereoisomers as exemplified by (*2R,3S*)-**10ab-ad** (3 examples) as the major products with >99% ee values (now Fig. 6b, data shown in green);
- (c) the three-step one-pot ADH-19/ENE-101/ADH-19 cascade gave the opposite *anti*-diastereoisomers as exemplified by (*2R,3S*)-**10ac** and (*2R,3S*)-**10ad** (2 examples) as the major products with 89->99% ee values (now Fig. 7, data shown in green).

All the abovementioned data have been added into our revised manuscript, and the corresponding HPLC analysis has been added into our revised Supplementary Information accordingly.

Comment: Does the work support the conclusions and claims, or is additional evidence needed? In addition to conversion after 24 h a time course experiment would add significant information.

Response: A time course experiment on model substrate (*Z*)-**1aa** and (*E*)-**1aa** has been performed according to your suggestion, and the conversion was checked at 2 h, 4 h, 6 h, 8 h, 10 h, 12 h and 24 h. It turns out that the reaction of (*Z*)-**1aa** finished in 2 h, providing product (*S*)-**2aa** in 99%

conversion and >99% ee, while it took 24 h for the reaction of (*E*)-**1aa** to finish, providing product (*S*)-**2aa** in >99% conversion and 88% ee. The result indicates that substrate (*Z*)-**1aa** is more reactive than (*E*)-**1aa**, and we have put this conclusion as a comment into the revised manuscript. Detailed information about the time course experiment can now be found in Supplementary Table 3.

Besides, a time course experiment on substrate **5aa** proved that the reaction time of 1.5 h was just enough for the reaction to reach completion, providing product (*S*)-**7aa** in >99% conversion and 90% ee value (Supplementary Table 9). Prolonging the reaction time to 2 h (89% ee), 3 h (83% ee) or 6 h (72% ee) led to the sharp decrease in enantioselectivity because of the configurational instability and fast racemization of (*S*)-**7aa** by keto-enol tautomerization in KPBS (Supplementary Table 10).

Comment: Are there any flaws in the data analysis, interpretation and conclusions? Do these prohibit publication or require revision?

There are no flaws. But it would be nice to receive information on the stability of **7aj**.

Response: Thanks for the comment. We performed the time course experiment on ENE-101 biocatalysed reduction of **5aj** to study the configurational stability of **7aj**, and detected the conversions and ee values every 10 min. As you can see from Supplementary Table 12, the reaction itself is much less enantioselective (28% *c*, 37% *ee* at 10 min), and it is obvious that **7aj** is not configurationally stable and racemizes under standard conditions (>99% *c*, 18% *ee* at 1 h & >99% *c*, 12% *ee* at 2 h & >99% *c*, 0% *ee* at 4 h). We have added a comment “ α -phenylthio vinyl aldehyde (*Z*)-**5aj** could also be transformed into (*S*)-**7aj** but with poor ee values, likely due to the low enantioselectivity of the reaction itself and the *in situ* racemization of the product (Supplementary Table 12)^{41,47,48}” in our revised manuscript.

In addition, optically active 2-phenylthio aldehydes such as **7aj** have been proved to be configurationally unstable by different research groups (ref. 41, 47, 48) including our group (ref. 41). For your information, we reported the dynamic kinetic resolution of racemic 2-phenylthio aldehydes with ketoreductases in our previous work by taking advantage of the configurational instability of 2-phenylthio aldehydes (ref. 41).

Comment: Is the methodology sound? Does the work meet the expected standards in your field?

The methodology is sound but it also falls short. All enzymes are used as cell free extract in mg and no activity assay is provided. This means that the decision to take one enzyme is always based on the coincidental activity in the cell free extract with might fluctuate considerably between batches. To correctly describe enzymatic reactions see Gardossi, L., Poulsen, P. B., Ballesteros, A., Hult, K., Svedas, V. K., Vasic-Racki, D. et al. (2010) Guidelines for reporting of biocatalytic reactions Trends in Biotechnology 28, 171-180 10.1016/j.tibtech.2010.01.001.

Response: Thank you for your constructive suggestion. We did carry out the activity assays of ENEs/GDHs/ADHs used in this work at the very beginning of this project. We have added this information and the activity assays of ENEs/GDHs/ADHs in Supplementary Table 1 and 2. The additional data provide the specific activity assay carried out for each enzyme and the activity data of each enzyme are shown in U/mg.

Comment: Is there enough detail provided in the methods for the work to be reproduced?

See comment above.

Response: Thanks for your comment. According to the constructive suggestions from you and the other two reviewers, we add more details including the protein sequence of the enzymes, activity assays of enzymes, detailed experimental procedures for all reactions, reaction optimization of chemoenzymatic cascades shown in Fig. 3, updated HPLC traces, *etc*, in our revised manuscript and supplementary information so that this work can be reproduced by other researchers.

On the other hand, we repeated the experiments for the synthesis of **2ak** (shown in Fig. 2, Fig. 3a and 3b) and **7af** (shown in Fig. 5a) during the revision as requested by Reviewer 2, as a result, almost exact same data of conversions, yields and ee values were obtained for all of them. Therefore, this work can be reproduced. We have put the comment “All data supporting the findings of this study are also available from the corresponding author upon request” in the Data Availability part in our revised manuscript.

REVIEWERS' COMMENTS

Reviewer #2 (Remarks to the Author):

In this revised manuscript Zhao et al. have addressed all issues raised by the previous reviewers (I have been reviewer #2) very diligently. The authors have added new experimental data and also complementary information, which has been requested by the reviewers. Therefore, the significance and experimental reproducibility of the publication has been improved.

In my point of view this manuscript is very strong and reports about a very useful new application of ene reductase-mediated biocatalysis providing an attractive solution for a relevant synthetic problem in organic synthesis.

I recommend the publication of this manuscript in Nat. Commun. as it is.

[Note from the Editor: Reviewer #2 was asked to assess also the response given to Reviewer #1]

Reviewer #3 (Remarks to the Author):

The authors have done an excellent job in improving the manuscript. The questions raised by all reviewers and in particular from my side have been fully answered. The manuscript can now be accepted.